# Genetic Analysis for Resistance to Sclerotinia Stem Rot, Yield and Its Component Traits in Indian Mustard [*Brassica juncea* (L.) Czern & Coss.]

**DOI:** 10.3390/plants11050671

**Published:** 2022-02-28

**Authors:** Manjeet Singh, Ram Avtar, Neeraj Kumar, Rakesh Punia, Ajay Pal, Nita Lakra, Nisha Kumari, Dalip Kumar, Anu Naruka, Mahavir Bishnoi, Rajbir Singh Khedwal, Raju Ram Choudhary, Anoop Singh, Ravindra Kumar Meena, Ankit Dhillon, Vivek K. Singh

**Affiliations:** 1Department of Genetics and Plant Breeding, CCS Haryana Agricultural University, Hisar 125004, India; ramavtar0706@gmail.com (R.A.); neerajkummar8@gmail.com (N.K.); punia.rakesh98@gmail.com (R.P.); nishaahlawat211@gmail.com (N.K.); dilipshroff@rediffmail.com (D.K.); anunaruka8@gmail.com (A.N.); mahaveer.bishnoi9@gmail.com (M.B.); rajbirsinghkhedwal1524@gmail.com (R.S.K.); rajuramchoudhary33@gmail.com (R.R.C.); ravimeena101295@gmail.com (R.K.M.); ankitdhill@gmail.com (A.D.); 2Department of Biochemistry, CCS Haryana Agricultural University, Hisar 125004, India; ajaydrdo@rediffmail.com; 3Department of Molecular Biology, Biotechnology and Bioinformatics, CCS Haryana Agricultural University, Hisar 125004, India; nitahaubotany2019@gmail.com; 4Department of Botany, Maharshi Dayanand University, Rohtak 124001, India; anoopnehra93@gmail.com

**Keywords:** Sclerotinia stem rot, gene action, combining ability, heterosis breeding, selection criteria

## Abstract

Understanding the mode of gene action that controls seed yield and Sclerotinia stem rot resistance in Indian mustard is critical for boosting yield potential. In a line × tester mating design, ten susceptible lines and four resistant testers were used to conduct genetic analysis. The significance of general combining ability (GCA) and specific combining ability (SCA) variances revealed that both additive and non-additive gene actions were involved in the inheritance of Sclerotinia stem rot resistance and yield attributing traits. In addition to 1000-seed weight and number of primary and secondary branches/plant, the genotypes RH 1569 (line) and DRMR 2035 (tester) appeared to be the strongest general combiners for Sclerotinia stem rot resistance. RH 1657 × EC 597317 was the only cross among several that demonstrated a significant desired SCA value for Sclerotinia rot resistance. Regarding SCA effects for yield and component traits, the cross RH 1658 × EC 597328 performed best, with a non-significant but acceptable negative SCA effect for resistance. DRMR 2035, RH 1222-28, RH 1569, RH 1599-41, RH 1657, RH 1658, and EC 597328 are promising genotypes to use as parents in future heterosis breeding and for obtaining populations with high yield potential and greater resistance to Sclerotinia stem rot disease in Indian mustard, based on GCA effects of parents, *per se* performance, and SCA effects of hybrids. Days to 50% flowering, number of primary branches/plant, main shoot length, and 1000-seed weight all had a high genotypic coefficient of variability (GCV), broad-sense heritability (h^2^bs), and genetic advance as percent of the mean (GAM) values, as well as significant and desirable correlations and direct effects on seed yield. As a result, these traits have been recognized as the most critical selection criterion for Indian mustard breeding programs.

## 1. Introduction

India is the world’s 4th largest grower and producer of oil-producing crops, accounting for ~19% of worldwide acreage and 2.7% of production. Oilseed crops, just after cereals, play an important role in the Indian agricultural economy. India is on track to become the world’s third-largest consumer market and an importer of edible oils, meeting over 60% of its domestic consumption through imports at the cost of USD 10 billion per year [1,2,3]. Domestic demand for edible oils and fats has been proliferating at 6% per year, but domestic output has only increased by 2% per year. The country’s significant scarcity of edible oils has been attributed to several issues, including the country’s ever-growing population, sudden climate change, rising household income, low productivity of oilseed crops, and a complicated disease–pest syndrome. Poor production performance of oilseed crops is the most important reason for India’s demand–supply mismatch in vegetable oils. Rapeseed–mustard is the third most extensively produced oilseed crop in India, accounting for ~32% of the country’s total oil pool [4].

Indian mustard [*Brassica juncea* (L.) Czern & Coss.] is the most widely cultivated oilseed crop in India, out of six economically important species of the rapeseed–mustard group, due to its greater sustainability to grow under diverse agro-climatic conditions [5,6,7]. Natural amphiploid (2n = 4x = 36, AABB) India mustard is developed by natural crossing and genome doubling between two diploid progenitors, *Brassica campestris* (2n = 2x = 20, AA) and *Brassica nigra* (2n = 2x = 16, BB) [8]. To meet the escalating demand for vegetable oils for India’s ever-increasing population, the productivity of Indian mustard must be increased. Even though this crop has achieved significant progress in terms of yield enhancement [4], current production is insufficient to meet the country’s demand. The requisite productivity goals can be met by producing high-yielding hybrids, which is possible in this crop due to abundant heterosis for seed yield and its components and a stable cytoplasmic male sterility/fertility restoration system [9,10,11,12,13,14,15,16,17]. Pure line-breeding procedures are also thought to reach the equilibrium point in yield enhancement since they do not produce enough genetic variability. In Indian mustard, hybrids, on the other hand, allow for a greater fraction of genetic variability and a more accessible high heterotic impact [18]. According to Sodhi et al. [19], heterosis breeding could be a viable option to pure line breeding for increasing Indian mustard yield potential, as it provides a yield advantage of 19–40% over the best pure line types. As a result, hybrids are one of the most viable alternatives for breaking the yield barriers in Indian mustard.

Indian mustard is exposed to various biotic and abiotic stresses that reduce and limit its output. As a result, in addition to boosting yield potential, the development of stress-tolerant/resistant cultivars is also critical to increase productivity. To achieve sustained and secure yield increase in Indian mustard, plant breeding focuses on crop cultivars with high yield potential and inbuilt resistance to critical yield-limiting factors [18,20]. Sclerotinia stem rot, caused by *Sclerotinia sclerotiorum* (Lib.) de Bary, is the most destructive fungal disease of Indian mustard at the moment, causing yield losses of 32–90% [6,21,22]. It also impacts the oil content (up to 35%) and quality [23]. *S. sclerotiorum* is a cosmopolitan and widespread phytopathogenic fungus with a broad host range that has gone from being inconsequential to symbolic due to global climatic changes and is currently one of the most devastating diseases of Indian mustard. Sclerotia, the survival structure of the pathogen, can survive in plant detritus for many years and act as the major inoculum for infection. It germinates myceliogenically (soil-borne infection) to produce mycelial hyphae that almost instantly invade the lower portion of plants, including the basal stem. In contrast, its carpogenic germination (airborne infection) has apothecia, which are cup-like structures with a 3–6 mm diameter that release ascospores to infect the upper sections of host plants. Almost all plant parts are affected, including cotyledons, leaves, branches, raceme, siliquae, and stems, with infected tissue displaying typical white fluffy cottony mycelial growth symptoms (Figure 1). On the other hand, infection of the stem causes girdling, which is linked to plant lodging and finally results in significant yield losses in Indian mustard [1,6,24].

Due to its broad infection ability and extended survival ability in the soil, proper treatment of Sclerotinia stem rot using cultural and chemical approaches is challenging, if not impossible. Furthermore, fungicide use is hazardous to the environment and increases the expense of crop production [6,25]. One of the most critical aspects of managing Sclerotinia stem rot would be resistant cultivars in Indian mustard [24]. However, a lack of suitable resistant sources has hampered breeding for resistance in the past. Previous attempts to uncover resistant sources against Sclerotinia stem rot failed miserably because all Indian mustard genotypes tested were sensitive/highly susceptible, and whatever resistant sources were declared were connected to wild and other Brassicaceae species [22,23,24,25,26,27,28,29,30,31,32,33,34]. However, in recent years, increased attention has led to identifying a few Indian mustard genotypes resistant to the disease [1,6,33,34,35].

Because Sclerotinia stem rot is one of the most critical constraints to Indian mustard production, cultivars with built-in resistance to this disease will be given even more importance to boost and sustain productivity and make this crop more profitable. Next to yield enhancement, breeding for disease resistance is essential in attaining optimal progress in edible oil production to satisfy future demands [18,20]. The success of any plant breeding program aimed at incorporating desirable traits, on the other hand, is entirely dependent on the availability of source material and understanding of genetic regulation of the trait(s) in question. As a result, crop breeders are constantly vigilant in determining desirable genetic traits to determine the most practical approach for breeding novel and elite cultivars [36]. Combining ability analysis frequently aids in selecting the best genotypic combinations for the development of superior hybrids [37].

Furthermore, plant breeders have a key difficulty in identifying the ideal parental combination to exploit heterosis in the F_1_ generation and produce superior transgressive segregants in the F_2_ and subsequent segregating generations in any hybridization program. A high *per se* performance genotype may not inevitably create better hybrids and/or transgressive segregants when employed in hybridization. Combining ability is a crucial notion that aids in selecting promising parents for hybridization and sheds light on the nature of gene actions that influence superior traits. Line × tester analysis is the most often used of the various mating designs available for combining ability and heterotic effect estimation with knowledge on the genetic control of metric traits in crop plants [38].

Furthermore, environmental or genotype × environmental interaction may cause variation across genotypes for several traits [39,40]. As a result, genotypes should be chosen based on their genetic rather than phenotypic characteristics [41]. Trait selection necessitates a thorough understanding of the nature and extent of genotypic variation and transmissibility and selection progress. The following selection indicators are commonly used to predict genetic gain under selection: genotypic coefficient of variation(GCV), phenotypic coefficient of variation (PCV), broad-sense heritability (h^2^bs), and genetic advance as percent of mean (GAM) [42,43]. Furthermore, knowledge of the interrelationship between resistance and yield and its components is critical for determining appropriate selection criteria for breeding Indian mustard for high yield potential and Sclerotinia stem rot resistance. As a result, correlation and path analysis is essential for designing an effective selection strategy and increasing the efficiency of breeding programs [44,45]. Plant breeders can use selection indices to fully utilize the response to selection for one or more characteristics. In reality, indices-based selection exhibits the response with direct selection and the correlated response because the selection is competent for other characters. Families from the base population should be assessed to derive genetic and phenotypic estimates such as h^2^bs, GAM, genetic correlation, and path coefficients for examining the collection of traits [46].

In this context, the current study was carried out to gather information about the nature and extent of gene action, combining ability effects and estimation of selection indices for Sclerotinia stem rot resistance, seed yield, and its component traits in Indian mustard.

## 2. Materials and Methods

### 2.1. Experimental Plant Materials and Crop Cultivation

The plant materials for this study consisted of 14 different genotypes of Indian mustard chosen for their different responses to Sclerotinia stem rot, seed yield, and its component traits (Table 1). The parental genotypes chosen as lines (females) are advanced lines with high yield potential developed at CCS HAU, Hisar, and have a narrow genetic base. In contrast, the parents chosen as testers (males) were obtained from various locations (DRMR Bharatpur and exotic collections from China and Australia) and have a broad genetic base. During the 2018–2019 Rabi season, 10 lines (susceptible genotypes) and 4 testers (resistant genotypes) were used in a Line × Tester mating design. As a result, the experimental materials included 40 F_1_ hybrids, 14 parents (10 Lines and 4 Testers), and two standard checks (RH 0749 and RH 725). During the Rabi season of 2019–2020, these plant materials were tested in a Randomized Complete Block Design (RCBD) with three replications at the Research Farm of Oilseeds Section, Department of Genetics and Plant Breeding, CCS HAU, Hisar. The plots were paired rows of 4 m length with a 30 × 10 cm separation (row × plant). Except for any fungicidal treatment to reduce Sclerotinia stem rot, the entire recommended package and practices were followed to establish a healthy crop.The best environmental conditions for Sclerotinia stem rot epidemic development in Indian mustard include high relative humidity (>80%), a temperature range of 5 to 25 °C, and wet soil [1]. According to the weather data in Appendix A, the crop’s field conditions were favorable for disease growth.

### 2.2. Field Evaluation and Data Collection for Seed Yield and Its Component Traits

The seed yield and its component traits were recorded as follow: days to 50% flowering (number of days from sowing to when 50% of the plants had flowered) and days to maturity (number of days from sowing to when 80% of the plants had reached physiological maturity) both recorded on plot basis, while other phenotypic traits were recorded from ten randomly selected un-inoculated/healthy plants from the center of each row as follows: plant height (measured in cm from the base of the plant to tip of the main raceme), number of primary branches/plant (assessed branches derived from the base of the main stem), number of secondary branches/plant (assessed branches emerged from the primary branches), main shoot length (measured in cm from base of most top primary branch to the tip of the plant), number of siliquae on main shoot (siliquae borne on main raceme), siliqua length (measured in cm from the 9 individual siliqua/plant, three each from bottom-, middle-, and top-borne branches), number of seeds/siliqua (counted from the same siliquae used to estimate siliqua length), 1000-seed weight (grams), seed yield/plant (grams), and oil content (determined in % using Soxhlet apparatus method).

### 2.3. Sclerotinia Sclerotiorum Inoculum Preparation, Artificial Disease Inoculation, and Resistance Assessment

For the pure culture preparation, *S. sclerotiorum* sclerotia (Hisar isolate) were employed. These sclerotia were first precisely washed in double distilled water, then surface sterilized by soaking for 10–12 s in 0.1% sodium hypochlorite solution. Sclerotia were then gently rinsed 3–4 times with distilled water to remove any remaining germicide before being dehydrated on aseptic blotter paper. Finally, these sclerotia were cut in half using surgical blades and aseptically transferred to Petri-plates containing Potato Dextrose Agar (PDA). These plates were kept in a BOD incubator at 22 ± 1 °C for five days. Using the techniques provided by Li et al. [26] and Singh et al. [1] a five-day-old pure culture of *S. sclerotiorum* was used to inoculate the main stems of 10 randomly chosen and labeled plants (other than the 10 representative plants selected for the assessment of yield and its component traits)from each genotype/replication. Mycelial discs (5 mm^2^) were cut from the borders of pure culture plates, placed on a paraffin wax strip (together with a moist cotton swab) and wrapped tightly around the main stem, as in our prior investigations [1,6]. Lesion length (cm) was measured on each infected plant using a linear ruler at 20 days after inoculation for each genotype/replicate, and the average was taken. According to the scale proposed by Garg et al. [47], genotypes were classified as highly resistant [mean lesion length (MLL) ≤ 2.5 cm], resistant (2.6–5.0 cm), moderately resistant (5.1–7.5 cm), susceptible (7.6–10.0 cm), and highly susceptible (>10.0 cm) based on MLL.

Based on the lesion length (cm) recorded from each plant, the following parameters were computed to determine genotypic response to Sclerotinia stem rot.

#### 2.3.1. Mean Lesion Length (cm)

*Mean lesion length* (cm) was calculated by averaging the *lesion length* (cm) measured from the inoculated main stems among all the tested plants by using the following formula:$$Mean\;lesion\;length\;\left(\mathrm{cm}\right)=\frac{\sum_{i=1}^{n}Lesion\;length\;\left(\mathrm{cm}\right)}{Total\;number\;of\;plants\;tested\left(n\right)}$$

#### 2.3.2. Disease Severity Index (DSI)

Disease severity index was computed by the following equation described by Ooi [48]:$$DSI=\frac{\sum \left(No.\;of\;plants\;in\;specific\;scale\times disease\;scale\right)}{Total\;no.\;\;of\;plants\;tested}$$
$$DSI=\frac{\sum \left(n\times 0\right)+\left(n\times 1\right)+\left(n\times 2\right)+\left(n\times 3\right)+\left(n\times 4\right)}{Total\;no.\;\;of\;plants\;\left(n\right)}$$

### 2.4. Statistical Analysis

The line × tester analysis procedure was utilized to estimate general and specific combining ability effects and variances, as outlined by Kempthorne [49] and elaborated by Singh and Chaudhary [50]. The genotypic coefficient of variability (GCV), phenotypic coefficient of variability (PCV), broad-sense heritability (h^2^bs), genetic advance (GA), and genetic advance as percent of the mean (GAM) were calculated according to Singh and Chaudhary [50]. Genotypic and phenotypic correlations between Sclerotinia stem rot resistance assessment parameters and seed yield and its component traits were evaluated as per Johnson et al. [51]. Seed yield/plant was used as a response variable, while its component traits were used as a causative variable, and path coefficients were calculated using genotypic correlation coefficients [52]. The statistical analysis was performed using the computer program Windowstat 8.0 (INDOSTAT Services Ltd., Hyderabad, India).

## 3. Results

### 3.1. Analysis of Variance for Line × Tester Analysis

Total variance from the combined ANOVA of Line × Tester was divided into variances owing to parents, lines (females), testers (males), females vs. males (lines vs. testers), crosses, parents vs. crosses, and GCA (general combining ability) and SCA (special combining ability) effects (Table 2). Parents, crosses (except days to maturity), and parents vs. crosses (except days to maturity, plant height, number of secondary branches per plant, and MLL) all exhibited significant variation in all of the traits evaluated in this study. There was also a significant difference in terms of variance owing to all source components of ANOVA for traits such as days to 50% flowering, main shoot length, number of seeds/siliqua, oil content, MLL, and DSI. Except for variation owing to testers and line vs.testers, the characters number of primary branches/plant, number of siliquae on main shoot, siliqua length, and 1000-seed weight showed significant variation for all components of the source of variation. Table 2 shows that the mean squares attributable to GCA effects (lines) for the characters days to 50% flowering, days to maturity, plant height, siliqua length, 1000-seed weight, mean lesion length, and disease severity index was very significant. The number of siliquae on the main shoot, 1000-seed weight, MLL, and DSI were all determined to have significant mean squares attributable to GCA effects (testers). Except for days to maturity and plant height, mean squares owing to SCA effects were highly significant (*p* ≤ 0.01).

### 3.2. Mean Performance of Parents and F_1_ Hybrids

Table 3 shows the mean performance of the parent genotypes, F_1_ hybrids, and two standard checks (commercial cultivars) in Sclerotinia stem rot resistance, seed yield, and its attributes. For the majority of the traits tested, all genotypes exhibited significant variation. Days to 50% flowering ranged from 35.0 to 53.0 days, with a mean of 42.7 days, whereas days to maturity have minimum, maximum, and mean values of 137.0, 157.0, and 150.0 days, respectively. The average plant height was 205.5 cm, with a range of 149.5 to 232.0 cm. Number of secondary branches/plant, main shoot length, number of siliquae on the main shoot, siliqua length, number of seeds/siliqua, and 1000-seed weight, among other yield parameters, showed wide variation that may be exploited in a varietal development program. Seed yield varied significantly between genotypes, ranging from 15.6 to 31.4 g/plant. The F_1_ hybrids viz., RH 1599-44 × DRMR 2035, RH 1599-44 × EC 597317, RH 1658 × EC 597328, and RH 1664 RH × 1222-28 exhibited increased yield potential due to superior performance for traits such as the number of primary and secondary branches/plants, main shoot length, and number of seeds/siliqua. Aside from that, these F_1_ hybrids had relatively high oil content. The MLL and DSI for Sclerotinia stem rot resistance ranged from 2.28 to 13.96 cm and 0.50 to 3.83, respectively. The parental genotype DRMR 2035 displayed a highly resistant response (MLL < 2.5 cm), whereas RH 1222-28 and EC 597328 exhibited a resistant response (MLL > 2.6 but <5.0 cm). However, most of the genotypes with high resistance were low yielders. Although, it is notable that the F_1_ hybrid RH 1569 × DRMR 2035 not only outperformed the best-released cultivar (RH 725) in terms of seed and oil yield (SYP = 29.00 g/plant; OC = 40.03%) but was also reasonably resistant to Sclerotinia stem rot (Figure 2). This combination can improve Sclerotinia stem resistance and seed and oil yield.

### 3.3. General Combing Ability (GCA) Effects of Lines and Testers

Table 4 shows the general combining ability effects of lines and testers. Lines RH 1599-41 and RH 1657 had significant (*p* ≤ 0.01) and negative GCA effects for both days to 50% flowering and days to maturity. Only the genotype RH 1569 had a significant and positive GCA effect for the number of primary and secondary branches/plant among all the lines and testers. Lines RH 1658 and PM 26 and testers EC 597328 and EC 597317 showed significantly positive GCA effects on main shoot length. The genotypes RH 1569 (line) and EC 597328 (tester) had a significantly positive GCA effect for number of siliquae on main shoot. Two lines, RH 1633 and RH 1658, and one tester, RH 1222-28, showed significant and positive GCA effects on siliqua length. For 1000-seed weight, lines RH 1569, RH 1633, and RH 1658, as well as testers RH 1222-28, DRMR 2035, and EC 597317, were good general combiners, while lines RH 1599-41 and PM 26 were good general combiners for number of seeds/siliqua. Only the parental lines RH 1569 and RH 1599-44 demonstrated a significantly positive GCA effect on seed yield/plant. The GCA effects for oil content were significant and positive for the lines RH 1566, RH 1599-44, RH 1658, PM 26, and tester DRMR 2035. The female parents, RH 1569, RH 1599-41, RH 1657, and the male parents, RH 1222-28 and DRMR 2035 had significant and desirable negative GCA impacts on MLL and DSI. These parental genotypes were effective Sclerotinia stem rot resistance combiners.

### 3.4. Specific Combining Ability (SCA) Effects of Crosses

Table 5 shows the SCA effects of 40 different cross combinations. Only the cross RH 1599-44 × EC 597317 showed significantly negative SCA effects for both days to 50% flowering and days to maturity among the 40 F_1_s studied, and were crucial for earliness in Indian mustard. RH 1658 × EC 597328 and RH 1899-53 × DRMR 2035 had significant (*p* ≤ 0.05) and positive SCA effects for both the number of primary and secondary branches/plants, while RH 1657 × DRMR 2035 had significant (*p* ≤ 0.05) and positive SCA effects for main shoot length and number of siliquae on main shoot. Four cross combinations, RH 1569 EC × 597317, RH 1599-41 × EC 597317, RH 1633 × EC 597328, and RH 1664 × DRMR 2035, showed significant (*p* ≤ 0.05) and positive SCA effects on siliqua length. In six and eight distinct crossings, respectively, significant (*p* ≤ 0.05) and positive SCA effects for the number of seeds/siliqua and 1000-seedweightwas detected. For seed yield/plant, positive SCA crosses are preferred. However, only one cross combination, RH 1658 × EC 597328, demonstrated a significant and positive SCA effect (5.19 for this trait). Sixteen out of forty cross combinations showed significant (*p* ≤ 0.05) and positive SCA effects for oil content. For Sclerotinia stem rot resistance assessment parameters such as MLL and DSI, the cross RH 1657 × EC 597317 showed significantly negative and desirable SCA effect. In contrast, three crosses, RH 1599-41 × EC 597328, RH 1657 × DRMR 2035, and RH 1664 × EC 597317, showed significantly positive and undesirable SCA effects.

### 3.5. Genetic Parameters and Selection Indices for Sclerotinia Stem Rot Resistance, Yield, and Its Component Traits

Table 6 shows the proportional contribution of lines, testers, and their interactions to total variance, genotypic and phenotypic coefficients of variation, heritability, and GA. In general, lines outperformed testers for every character, except main shoot length (21.34%) and number of siliquae on main shoot (31.64%), where testers outperformed the lines. The contribution of lines was highest for the character, day to 50% flowering (79.15%), followed by plant height (61.48%), DSI (58.45%) and MLL (47.14%), while it was lowest for main shoot length (13.70%). Testers contributed the most to the number of siliquae on the main shoot (31.64%), 1000-seed weight (26.10%), and main shoot length (21.34%), but the least to seed yield/plant (1.49%). The number of seeds/siliqua (70.89%) had the highest line × tester interaction, whereas days to 50% flowering had the lowest (18.03%). For the characters, number of primary and secondary branches/plant, main shoot length, number of siliquae on main shoot, siliqua length, number of seeds/siliqua, seed yield/plant, and oil content, the SCA variances were higher than the GCA variances. In contrast, the GCA variance was high for the other traits. The SCA variances were significant (*p* ≤ 0.05), with the main shoot length (31.87) being the largest, followed by the number of siliquae on the main shoot (15.68), and the days to maturity (0.07) and DSI (0.07) being the lowest. Plant height (13.66) had the highest GCA variance, whereas oil content had the lowest (0.05). The proportions of σ^2^ GCA/σ^2^ SCA were less than unity (<1) for characters such as number of primary and secondary branches/plant, main shoot length, number of siliquae on main shoot, siliqua length, number of seeds/siliqua, seed yield/plant, and oil content, but were greater than unity (>1) for the remaining characters such as days to 50% flowering, days to maturity, plant height, 1000-seed weight, MLL, and DSI. All examined traits had only marginally larger PCV than GCV, except for seed yield/plant, which had a 1.5-fold higher PCV than GCV value. The GCV and PCV of all the characters in this experiment ranged from low to high. Oil content had the lowest GCV and PCV values (1.48% GCV and 1.51% PCV), whereas MLL had the greatest (25.39% GCV and 31.48% PCV). Estimates of broad-sense heritability (h^2^bs) ranged from very low (25.31% for days to maturity) to very high (95.55% for oil content). Days to 50% flowering, 1000-seed weight and oil content had high heritability estimates (>80%), while the number of primary and secondary branches/plant, main shoot length, number of siliquae on main shoot, siliqua length, MLL, and DSI had moderately high heritability estimates (60–79%). Days to maturity and seed yield/plant have low heritability estimates (<40%). GAM for days to maturity and DSI ranged from 2.15 to 46.14. For DSI, MLL, 1000-seed weight and days to 50% flowering, GAM was relatively high (>20%).

### 3.6. Correlation and Path Analysis

Table 7 shows the phenotypic and genotypic correlation coefficients between pairs of traits. Most of the examined variables had significant phenotypic and genotypic correlations. There was close agreement between the two; nevertheless, the differences were prominent in some cases, implying that environmental factors play a significant role in estimating these parameters. Days to maturity, number of primary and secondary branches/plant, main shoot length, number of siliquae on main shoot, siliqua length, number of seeds/siliqua, and 1000-seed weight were all significantly and positively correlated with seed yield at both genotypic and phenotypic levels. In contrast, days to 50% flowering and DSI had a significant negative association with seed yield at the genotypic level. Except for main shoot length, none were significantly correlated with oil content, and seed yield and oil content were not connected to one another or to MLL. Still, both resistance-related measures, namely, MLL and DSI, had a positive correlation at both genotypic and phenotypic levels.

Furthermore, the DSI was negatively associated with days to 50% flowering, plant height, and the number of primary and secondary branches per plant. In contrast, it was positively associated with siliqua length, number of seeds/siliqua, and 1000-seed weight. It revealed that comparatively late and tall genotypes required more days to reach 50% flowering, producing more primary branches/plant and siliquae on the main shoot, but with shorter main shoot lengths and fewer seeds/siliqua. The genetic correlation was subsequently investigated using the path coefficient analysis approach, which partitioned the correlation coefficient into direct and indirect effects via alternative traits. The resultant variable was seed yield/plant, which was the complex outcome of various characters, whereas its component traits were treated as causal variables. Table 8 shows the direct and indirect effects of these characters. The strongest direct effect (2.645) was from day to 50% flowering, which had a substantial negative genotypic correlation with seed yield (−0.165). The positive direct effects of days to 50% flowering were partially or entirely offset by the negative indirect effects of the number of seeds/siliqua, main shoot length, DSI, and siliqua length. The least direct influence (0.034) was on the number of secondary branches/plant, which had a highly significant positive connection with seed yield (0.278). Seed yield was directly affected by 1000-seed weight (2.525), plant height (1.642), main shoot length (1.430), number of seeds/siliqua (1.349), and number of primary branches/plant (1.197). Days to maturity (−3.753) and siliqua length (−2.092) had the most negative direct effect on yield. Furthermore, siliqua length influenced seed yield negatively via MLL and DSI.

## 4. Discussion

Breeding cultivars with high yield potential and resistance to major diseases is a primary goal of crop improvement projects. When hybridization is attempted in a specific mating design, identifying the best parental genotype combinations allows breeders to take advantage of their heterotic effects and shows that superior transgressive segregants are available in the F_2_ and subsequent segregating generations of that cross. The ability to examine the combining ability and heterotic effects in selecting superior parents for the future requires the mean values of parents and F_1_ combinations. Line × Tester analysis is the best way to examine the potentiality of contrasting lines (females) and testers (males) for their combining ability and gene action for different traits [38]. Based on their *per se* performance and combining ability effects, the current study was conducted to identify the best parental genotypes and their cross combinations for Sclerotinia stem rot resistance, seed yield, and component traits. The data from the traits analyzed can establish a helpful breeding strategy for future high-yielding Indian mustard hybrids/varieties with increased Sclerotinia stem rot resistance.

There must be sufficient genetic variation within the selected lines for any crop breeding program to succeed. The mean square due to genotypes for agronomic traits, MLL, and DSI was highly significant (*p* ≤ 0.01) in the analysis of variance (ANOVA), showing that there was a lot of genotypic variation for these characters among the genotypes tested. As a result, line × tester analysis can split total genetic variance into its appropriate orthogonal components. For most of the traits tested, ANOVA for line × tester mating design demonstrated the significance of their orthogonal components, namely, parents, crosses, and parent vs. Crosses. This implies a high level of genetic variability in both males (testers) and females (lines) parents and their F_1_ hybrids, allowing for a more in-depth investigation of genetic variation by combining ability analysis and the measurement of the extent heterosis for these traits. Kaur et al. [53], Meena et al. [54,55,56], Vaghela et al. [57], Chaudhary et al. [58], and Gupta et al. [9] observed high genetic variability for yield and its component traits in Indian mustard. Godoy et al. [59], Castano et al. [60], Achbani et al. [61], and Grecizes-Besset et al. [62] in sunflower and Ferreira et al. [63] in common bean revealed high genetic variability for resistance/susceptibility to *S. sclerotiorum*.

Combining ability analysis is frequently used to compare parental performance and better understand the basis of gene action that causes trait manifestation. Furthermore, combining ability is often helpful in forecasting the heterotic response of specific lines/genotypes in various cross combinations and acquiring superior transgressive segregants in the F_2_ and subsequent segregating generations. The GCA effect is used to select desirable parents, while the SCA effect is used to evaluate testcross progenies to form heterotic hybrids [37,38,64]. For days to flowering, siliqua length, and 1000-seed weight, significant mean squares attributable to lines and/or testers (GCA) effects and line × tester (SCA) effects show an interplay of additive and non-additive gene effects for the expression of these traits. While additive genetic action influenced the inheritance of days to maturity and plant height, non-additive gene action was significant for expressing the rest of the traits, as revealed by significant mean square due to lines, testers (GCA) and line × tester (SCA) effects for these traits, respectively. Both additive and non-additive genetic effects influenced Sclerotinia stem rot resistance, as evidenced by significant mean squares of lines, testers, and line × tester interactions for both resistance evaluation criteria, namely, MLL and DSI. These findings are consistent with those of Khan et al. [32], Disi et al. [30], Godoy et al. [59], Castano et al. [60], and Achbani et al. [61], who found that both additive and non-additive genetic action influenced Sclerotinia rot resistance inheritance and could be improved using the recurrent selection procedure. The significant effect of GCA on the sum of squares of SCA suggested that early generation selection of resistant progenies could be successful.

When utilized in hybridization, selecting parents based on their *per se* performance may not always be a fair method because a phenotypically worthy parent may not always produce superior hybrids and transgressive segregants in the segregating generations. As a result, it is necessary to choose parents based on their genetic assets. The parents significant GCA effects are primarily due to their additive and additive × additive gene effects, a fixable component in segregating generations. Based on their GCA effects, parents should be chosen for hybridization to isolate superior segregants in the F_2_ and following generations [38,63,64]. Our findings revealed that none of the parents were good general combiners for all of the traits investigated. This conclusion suggests that collective breeding strategies with optimal mating designs must accumulate desirable alleles into a single genetic background. Higher negative GCA values offered better resistance to Sclerotinia stem rot, while higher positive GCA values indicated increased susceptibility. The line RH 1599-41 was involved in three of the top four Sclerotinia stem rot-resistant and early flowering hybrids and had the highest negative GCA effect for DSI and early maturity. Regrettably, it proved to be a poor combiner in seed yield per plant. Positive combining ability and heterotic effects are beneficial for yield component traits like number of primary and secondary branches/plant, seeds/siliqua, siliqua length, and 1000-seed weight because they provide potential for improving yield. Despite ranking second in general effects for yield, the line RH 1569 may be the best choice because it has a good GCA effect for the majority of important yield component traits such as number of primary and secondary branches/plant, number of siliquae on main shoot, and 1000-seed weight, while also having the best general effects for Sclerotinia stem rot resistance. The best general combiners for resistance and the two most crucial yield-related qualities, 1000-seed weight and number of major branches/plant, were DRMR 2035 and RH 1222-28 among testers. Short plant stature and vegetative period are also essential to creating lodging tolerant and comparatively large seed filling period cultivars of Indian mustard for yield and Sclerotinia stem rot resistance. While early maturity allows enough time to raise the following crop, late maturity reduces yield and oil quality due to increased temperature during the final stages of the crop [65]. Negative combining ability and heterotic effects are thus required for these traits. PM 26 and RH 1658 are good general combiners for shortening vegetative development and reducing plant height.

Overall, the genotypes RH 1569 (line) and DRMR 2035 (tester) looked to be the strongest general combiners for Sclerotinia stem rot resistance and most yield component traits. They should do so well in hybrid combinations with other parents. The greatest criterion for maximizing heterosis in F_1_ hybrids is to choose parents based on their SCA values. Negative SCA crosses for days to flowering, maturity, and plant height were wanted, while positive SCA crosses for other yield-related attributes were desired.

Contrary to what was expected based on the parent’s GCA, significant SCA effects in the desired direction demonstrate positive deflections with regards to the F_1_ crosses. The SCA effect, which considers loci with non-additive and epistatic gene effects, can also identify high heterotic F_1_ hybrids. Negative SCA effects for MLL and DSI are desired because they lead to resistance, whereas positive SCA adds to Sclerotinia stem rot susceptibility. RH 1657 × EC 597317 was the only cross that demonstrated a significant desired SCA value for MLL and DSI among other cross combinations, showing the involvement of a particular effect in this hybrid’s resistance expression. It could be because of the good general combiner RH 1657, which has additive effects, and the lousy combiner parent (EC 597317), which has epistatic effects. However, it revealed adverse SCA effects for the majority of yield-related traits. All other crossings, except for cross RH 1657 × EC 597317, had unacceptable and/or insignificant SCA effects for both resistance evaluation criteria, demonstrating that genes/alleles giving Sclerotinia stem rot resistance are recessive over susceptibility. Furthermore, crosses involving both parents with significant GCA effects for resistance had poorer SCA effects, implying the existence of a complex non-allelic gene interaction for resistance and/or that both of these parents may have identical resistance alleles thus could not benefit from fixable gene effects. Similarly, Van Becelaere and Miller [66] found that GCA effects of both male and female lines were crucial for Sclerotinia head rot resistance in sunflower, but SCA effects were not significant. The ranking of hybrids for resistance assessment parameters in terms of mean values and SCA effects revealed that the lowest mean values did not always predict significant adverse SCA effects, and vice versa. Ross et al. [67] found a pattern of combining ability effects in grain sorghum, as did Satyanarayana [68] in rice. Most of the hybrids had insignificant SCA effects for both seed yield and its component traits. The hybrid RH 1658 × EC 597328, on the other hand, had the best SCA effects for yield and components, as well as an insignificant but desirable negative SCA effect for resistance. This cross can develop hybrids or transgressive segregants with excellent seed and oil yields and resistance to Sclerotinia stem rot. The cross RH 1599-44 × EC 597317 showed significant SCA effects for lowering plant height and vegetative and maturity periods in this study. As a result, including the parents in a specific mating design such as diallel or triallel may increase the possibility of producing high-yielding, resistant segregants and developing hybrids. The inconsistency of GCA and SCA effects suggests that these traits have complex gene connections.

The contribution to the total variation of the lines, testers, and their interactions support prior results that general effects were more relevant than specific effects for Sclerotinia rot resistance [59,61,62]. On the other hand, specific effects had a greater impact on yield component traits and oil content [9,11,54,57]. GCA and SCA variances revealed the role of both additive and non-additive gene action in the expression of most examined traits. The significance of SCA in creating heterotic crosses for most yield-related traits was highlighted by higher SCA variance than GCA variance of lines and testers. The allele frequencies between parental genotypes determine the magnitude of the GCA/SCA variance ratio. For most of the yield-attributing traits, such as number of primary and secondary branches/plant, main shoot length, number of siliquae on main shoot, siliqua length, number of seeds/siliqua, and oil content, this ratio revealed the predominance of non-additive gene action (SCA variance). This indicates that selecting superior plants for these traits should be deferred to subsequent generations. Although additive gene action (GCA variance) was essential for days to flowering, days to maturity, plant height, and Sclerotinia rot resistance, selection of phenotypically superior plants would be effective in early generations. These findings are consistent with those of Meena et al. [54], Vaghela et al. [57], and Gupta et al. [69], but differ from those of Dahiya et al. [70] and Chaudhary et al. [58]. Because both additive and non-additive gene actions were necessary for the inheritance of the studied traits, hybridization methods that simultaneously use additive and non-additive gene effects, such as diallel selective mating scheme or reciprocal recurrent crossing, could be helpful in the genetic improvement of the traits under consideration.

The current study’s PCV values were larger than the GCV values for all traits, showing that environmental effects on the characteristics were more relevant than genotypic effects. Previous publications on higher PCV values than GCV values for yield and component traits in Indian mustard [71,72,73,74,75] complement our findings. Seed yield/plant had a considerable discrepancy for PCV and GCV, owing to the greater confounding effect of environment on this characteristic. Furthermore, due to the concealing effect of the environment, GCV and PCV estimates do not substantiate the exhaustive extent of heritable variation, which may be assessed more precisely with heritability and genetic advance estimates [76]. Heritability and genetic advancement of a particular trait determine genetic gain or responsiveness to selection. Singh [77] divided broad-sense heritability estimates (h^2^bs) into four categories: low (<40%), moderate (40–59%), moderately high (60–79%), and extremely high (>80%). Days to 50% flowering, 1000-seed weight, oil content, number of primary and secondary branches/plant, main shoot length, number of siliquae on main shoot, siliqua length, MLL, and DSI all had moderately high to extremely high heritability values in this study. If these traits are employed to boost seed and oil yield and Sclerotinia stems rot resistance in Indian mustard, the projected gain from the selection will be considerable. The high heritability estimates also suggest that the environment poorly influences certain traits. Heritability estimates for plant height and number of seeds/siliqua were intermediate, indicating that these parameters’ environmental and genetic effects were indistinguishable. Days to maturity and seed yield/plant have low heritability estimates, and direct selection for these traits would be challenging due to the high environmental impact. As a result, component traits with high heritability and strong desirable correlations can be picked concurrently to select these traits. Several studies found moderate to high heritability for yield and its component traits in Indian mustard, similar to the current study [78,79,80,81,82,83]. The heritability of Sclerotinia stem rot resistance evaluation parameters (MLL and DSI) was moderate, indicating that early selection for Sclerotinia stem rot resistance may be successful. Sclerotinia stem rot resistance has been estimated to be heritable by various researchers, with values ranging from 21 to 88% [32,84,85]. Genetic advance as percent of mean (GAM) values are divided into three categories: low (0–10%), moderate (10–20%), and high (>20%) [86]. High GAM values for DSI, MLL, 1000-seed weight, and days to 50% flowering imply that these traits are highly likely to be improved through selection. The rest of the traits have intermediate to low GAM values, indicating that selection in the early generation is ineffective in improving these traits. High heritability combined with moderate to high GAM indicates that component traits are primarily driven by genes and are only marginally influenced by the environment, making them easily exploitable for seed yield improvement [87]. Selection is difficult for a character with low to moderate h^2^bs and low GAM [88]. Selection for high-yielding genotypes with inbuilt resistance to Sclerotinia stem rot should be delayed to confirm the homozygosity of genes controlling resistance and yield and its component traits. It should be based on a high number of primary and secondary branches/plant, longer main shoot length, high number of siliquae on main shoot, more 1000-seed weight, fewer days to 50% flowering, and low MLL and DSI under multiple locations and years.

For most of the traits tested, the genotypic correlation was higher than the phenotypic correlation, showing genetic relationships. As a result, phenotype-based selection would work [82,89,90]. Seed yield in mustard is determined by several interdependent and environment-dependent traits. The number of primary and secondary branches, main shoot length, number of siliquae on main shoot, siliqua length, number of seeds/siliqua, and 1000-seed weight all had positive and significant genotypic relationships in the current study. This shows that seed yield will be improved by indirect selection through component traits. Similarly, in Indian mustard [78,91,92,93,94] a strong link between these characteristics and seed yield has been found. In addition, days to flowering showed a significant and negative relationship with seed yield, which is beneficial because early flowering in Indian mustard allowed for shorter vegetative and reproductive stages and increased grain filling time. However, because the development of high-yielding and early maturing varieties is the primary goal of mustard breeders, the significant and positive correlation between seed yield/plant and days to maturity is undesired. Indirect selection of yield through its component traits necessitates the examination of genotypic and phenotypic relationships among the traits. However, the correlation-based choice is ineffective because it only shows the linear relationship between two traits.

Path coefficient analysis has been widely used in crop breeding projects to discover the relationship between seed yield and component traits [87]. Days to 50% flowering, 1000-seed weight, number of primary branches/plant, and main shoot length exhibited very high and positive direct effects on seed yield/plant in this study, indicating that these traits should be prioritized for indirect selection. However, through the MLL and DSI, siliqua length negatively impacted seed yield. To generate high-yielding and disease-resistant cultivars, extensive genetic analysis and identification of alleles causing linkage drag among Sclerotinia stem rot resistance and seed yield are necessary. Through siliqua length, 1000-seed weight had the greatest indirect effect on seed yield, implying that longer siliquae are more likely to have higher seed weight. Our findings are supported by Lodhi et al. [95], who used path analysis to examine 90 different Indian mustard genotypes and found that main shoot length, number of primary branches/plant, and number of seeds/siliqua had the most significant direct effect on seed yield. Pandey et al. [96] found that plant height, 1000-seed weight, number of seed/siliqua, siliqua on main raceme, and primary branches/plant had the greatest direct effect on seed yield. Saroj et al. [83] found that total seed yield had the highest positive direct effect, followed by siliquae on the main shoot and seed size, in a separate study involving 289 diverse Indian mustard accessions analyzed over two seasons.

## 5. Conclusions

This study highlighted the rewarding parents and crosses of Indian mustard that can be exploited to launch effective breeding strategies for developing high-yielding and Sclerotinia stem rot-resistant cultivars. Based on results, it is concluded that significant and desirable GCA effects were shown by DRMR 2035 among the testers and RH 1569 among the lines that were considered to be good general combiners for Sclerotinia stem rot resistance, seed yield, and oil content, as well as for most of the yield attributing traits. The cross combination of RH 1657 × EC 597317 and RH 1658 × EC 597328 showed excellent SCA performance for the resistance and yield contributing traits. These parents and hybrids could be utilized as good combiners and isolated to obtain desirable segregate for combining high resistance levels to Sclerotinia stem rot with superior seed yield and oil content in the future Indian mustard breeding program. This study was suggested for a breeding program for which the advantages of both types of variances, namely, additive and non-additive, are chosen. Days to 50% flowering, number of primary branches/plant, main shoot length, and 1000-seed weight and disease severity index are the best selection criterion in the field for use in breeding programs for simultaneous improvement in yield and resistance because of its high heritability and desirable association with seed yield.

## Figures and Tables

**Figure 1 plants-11-00671-f001:**
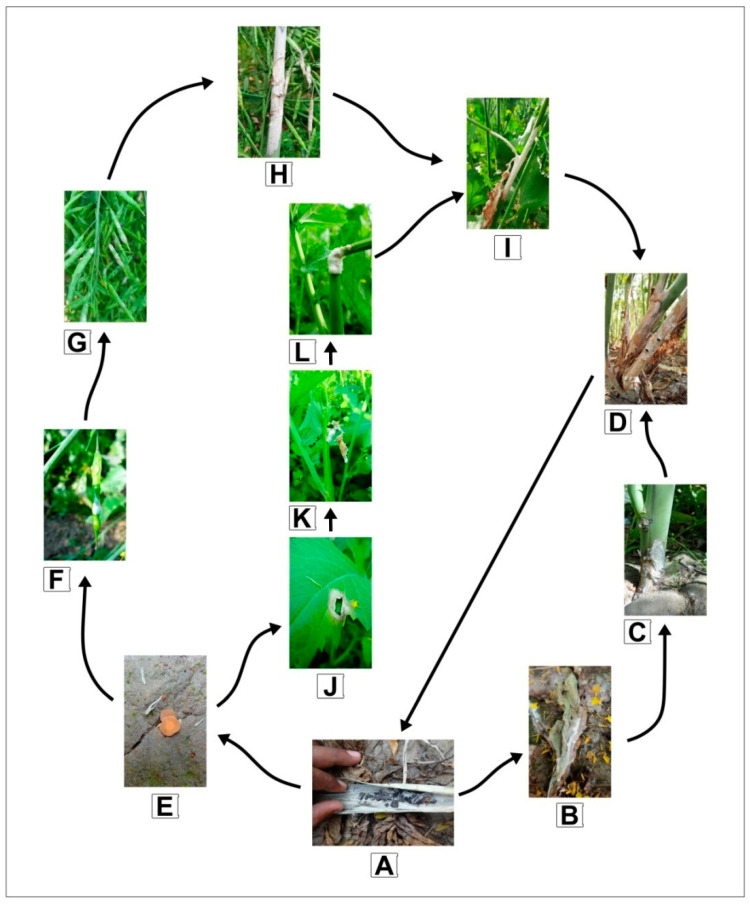
*Sclerotinia sclerotiorum* life cycle and complex mode of infection in Indian mustard. (**A**) Sclerotia in the Indian mustard plant debris. (**B**) Myceliogenic germination of sclerotia resulting in actively growing mycelia on dead leaves. (**C**) Initiation of myceliogenic infection on basal stem with characteristic symptoms: water-soaked lesions with fluffy, white mycelium. (**D**) Highly infected stem with bleached and necrotic tissues that finally expand blotches of fluffy white mycelium, usually with black colored sclerotia. (**E**) Carpogenic germination of sclerotia leads to the formation of apothecia. (**F**) Apothecia release ascospores embedded on senescing petals and get stuck on healthy siliqua. (**G**) Infected siliquae with characteristic symptom of fluffy mycelial growth. (**H**) Spread of disease by plant-to-plant contact and infected siliquae spread inoculum on mustard raceme. (**I**) Infection spread further on plant raceme, (**J**) infected petal fall on leaf to initiate infection, infection spread further to cause (**K**) complete destruction of leaf and (**L**) stalk of mustard.

**Figure 2 plants-11-00671-f002:**
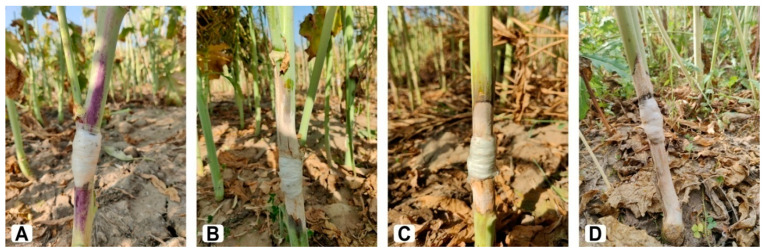
Sclerotinia stem rot response of the two parental Indian mustard genotypes, resistant DRMR 2035 (**A**) and susceptible RH 1569 (**B**), and their moderately resistant F_1_ hybrid RH 1569 × DRMR 2035 (**C**) in comparison with highly susceptible commercial cultivar RH 0749 (**D**).

**Table 1 plants-11-00671-t001:** List of parental lines used in the present investigation along with their pedigree, source/origin, and their response towards Sclerotinia stem rot.

Parents (Lines/Testers)	Genotype Code	Pedigree	Sources/Origin	Response to Sclerotinia Stem Rot	Reference
Lines	RH 1566	RH 0734/RH 0202	CCS HAU, Hisar/India	Susceptible	[6,35]
	RH 1569	RH 0735/RH 0119	CCS HAU, Hisar/India	Highly susceptible	[6,35]
	RH 1599-41	RH 0802/JM 18	CCS HAU, Hisar/India	Highly susceptible	[35]
	RH 1599-44	RH 0803/RH (00) 7003	CCS HAU, Hisar/India	Highly susceptible	[35]
	RH 1633	RB 50/RH 0555	CCS HAU, Hisar/India	Highly susceptible	[6,35]
	RH 1657	RH 0803/RH 0745	CCS HAU, Hisar/India	Susceptible	[35]
	RH 1658	RH 0804/RH 0745	CCS HAU, Hisar/India	Susceptible	[35]
	RH 1664	RH 0835/JM 18	CCS HAU, Hisar/India	Highly susceptible	[35]
	RH 1899-53	RH 1008/NPJ 153	CCS HAU, Hisar/India	Susceptible	[35]
	PM 26	VEJ Open/Pusa Agrani	IARI, New Delhi/India	Susceptible	[35]
Testers	RH 1222-28	RH 0406/RH 0401-B	CCS HAU, Hisar/India	Resistant	[6,33,34,35]
	DRMR 2035	PHR-1/BEC-107	DRMR, Bharatpur/India	Highly resistant	[6,33,35]
	EC 597328	Exotic	Exotic Collection/Chinese	Resistant	[6,33,35]
	EC 597317	Exotic	Exotic Collection/Australia	Moderately resistant	[33,35]

**Table 2 plants-11-00671-t002:** ANOVA of Line × Tester analysis for seed yield and its component traits and Sclerotinia stem rot resistance in Indian mustard.

Traits	Mean Squares									
	Genotypes	Parents (P)	Lines (L)	Testers (T)	L vs. T	Crosses (C)	P vs. C	Line Effects	Tester Effects	L × T Effects
DF	68.91 **	88.35 **	72.53 **	13.00 *	456.77 **	53.21 **	428.38 **	182.51 **	19.48	13.86 **
DM	56.39 **	99.42 **	115.94 **	50.31	98.12	43.49	0.01	95.82 **	49.83	25.34
PH	525.29 **	1464.21 **	1549.49 **	237.79	4376.01 **	224.34 **	56.58	597.66 **	224.57	99.87
NPB	2.12 **	2.00 **	2.18 **	0.64	4.47 **	2.10 **	4.66 **	2.93	2.49	1.78 **
NSB	17.89 **	17.06 **	21.68 **	8.25 *	1.86	18.49 **	5.39	18.02	39.4	16.32 **
MSL	141.49 **	123.44 **	148.93 **	61.23 *	80.64 *	120.38 **	1199.74 **	71.46	333.89	112.96 **
NSMS	84.29 **	43.03 **	47.63 **	19.95	70.79 *	98.18 **	78.82 *	101.31	403.79 **	63.19 **
SL	0.49 **	0.50 **	0.58 **	0.40 **	0.02	0.33 **	6.77 **	0.70 **	0.21	0.21 **
NSPS	4.17 **	3.33 **	2.89 **	3.27 **	7.49 **	2.76 **	70.47 **	3.13	1.02	2.82 **
TSW	1.36 **	1.98 **	2.29 **	1.69 **	0.01	1.16 **	0.89 **	2.70 **	3.95 **	0.34 **
SYP	41.52 **	26.03 *	20.85	26.17	72.33 *	28.90 **	735.29 **	44.88	5.58	26.16
OC	1.03 **	1.52 **	1.62 **	1.62 **	0.25 **	0.89 **	0.35 **	1.39	0.76	0.74 **
MLL	18.94 **	38.11 **	17.23 **	7.97 *	316.47 **	12.83 **	8.15	35.12 **	19.91 *	4.61 **
DSI	1.52 **	3.12 **	0.85 **	0.91 **	30.13 **	0.99 **	1.21 *	2.52 **	1.57 *	0.42 **

** Significant at *p* ≤ 0.01; * Significant at *p* ≤ 0.05; DF-Days to Flowering (50%); DF-Days to flowering, DM—Days to maturity, PH—Plant height (cm), NPB—Number of primary branches/plant, NSB—Number of secondary branches/plant, MSL—Main shoot length (cm), NSMS—Number of siliquae on main shoot, SL—Siliqua length (cm), NSS—Number of seeds/siliqua, TSW—1000 seed weight (g), SYP—Seed yield/plant (g), OC—Oil content (%), MLL—Mean lesion length (cm), DSI—Disease severity index.

**Table 3 plants-11-00671-t003:** Mean performance of parents and their F_1_ hybrids for Sclerotinia stem rot resistance assessment parameters as well as seed yield and its component traits.

Genotypes	DF	DM	PH	NPB	NSB	MSL	NSMS	SL	NSS	TSW	SYP	OC	MLL	DSI
RH 1566	52.00	156.67	225.00	7.00	19.00	73.17	56.77	3.65	13.24	4.98	24.30	38.73	8.94	2.93
RH 1569	48.00	150.67	216.34	7.67	18.50	68.50	56.83	4.50	14.90	5.94	24.85	39.50	9.88	3.23
RH 1599-41	39.00	141.33	183.72	5.83	20.50	89.34	52.84	3.87	15.37	4.31	20.30	39.83	12.14	3.63
RH 1599-44	46.00	153.67	221.17	5.99	17.34	76.67	54.84	3.73	14.07	4.77	21.15	39.80	12.85	3.57
RH 1633	41.00	149.00	202.01	6.84	20.34	89.00	56.17	4.10	16.37	6.11	25.35	38.63	13.58	3.67
RH 1657	43.00	149.00	202.50	7.01	20.67	74.84	56.67	3.47	14.40	4.66	21.20	40.23	7.61	2.40
RH 1658	38.00	151.67	178.67	5.84	20.00	79.33	48.84	4.00	14.77	6.47	23.95	38.53	7.63	2.47
RH 1664	43.00	144.67	192.67	5.67	15.17	71.17	44.83	4.80	15.90	5.54	18.85	38.63	12.96	3.70
RH 1899-53	46.00	155.00	204.95	7.67	24.84	73.17	51.50	3.50	13.60	4.05	24.90	40.30	7.56	2.43
PM 26	36.00	137.00	149.45	5.33	16.50	79.67	52.00	3.60	14.34	4.07	18.45	38.50	10.16	3.13
RH 1222-28	51.00	157.00	222.22	7.00	17.67	78.67	54.67	4.33	15.14	6.16	21.60	38.77	3.72	1.20
DRMR 2035	50.00	150.33	232.00	7.50	21.67	76.84	59.67	3.47	12.60	4.81	15.60	38.70	2.28	0.50
EC 597328	48.00	154.00	212.58	6.66	19.50	73.83	53.83	3.97	13.60	4.42	18.60	38.73	4.97	1.47
EC 597317	53.00	147.67	214.17	7.67	20.17	68.34	55.84	4.10	13.70	4.96	21.90	40.20	6.06	1.80
RH 1566 × RH 1222-28	46.00	155.00	217.84	7.67	19.50	77.33	58.00	4.37	15.97	5.67	27.80	38.67	8.38	2.73
RH 1566 × DRMR 2035	45.00	155.33	210.84	7.17	20.00	77.50	56.00	4.14	15.83	5.41	27.30	40.07	8.15	2.63
RH 1566 × EC 597328	47.00	146.00	211.34	7.36	20.66	73.73	54.40	3.90	13.55	4.16	22.70	39.83	9.63	3.03
RH 1566 × EC 597317	50.00	157.00	209.00	5.17	15.19	84.00	57.17	3.87	14.60	5.43	25.70	39.93	7.70	2.47
RH 1569 × RH 1222-28	45.00	151.00	204.84	8.34	24.67	88.50	60.00	4.43	16.04	5.73	29.55	38.73	6.49	1.97
RH 1569 × DRMR 2035	43.00	151.67	218.67	7.83	21.34	80.67	58.34	4.37	16.37	5.48	29.00	40.03	5.67	1.57
RH 1569 × EC 597328	45.00	148.33	215.00	7.17	21.50	89.50	69.50	3.90	16.04	4.86	30.15	38.77	6.41	2.03
RH 1569 × EC 597317	51.00	151.67	213.83	8.82	22.00	78.17	56.50	5.33	14.94	5.84	26.35	38.83	8.52	2.80
RH 1599-41 × RH 1222-28	36.00	145.33	192.84	6.67	18.17	83.17	54.84	4.00	16.40	5.35	26.70	38.80	5.54	1.70
RH 1599-41 × DRMR 2035	36.00	142.00	203.67	6.17	16.50	79.17	56.17	3.93	15.77	4.60	24.10	38.83	6.16	1.77
RH 1599-41 × EC 597328	40.00	144.33	211.67	7.17	20.50	88.67	57.17	4.07	16.80	4.41	24.50	38.70	9.87	3.03
RH 1599-41 × EC 597317	36.00	149.00	207.67	6.19	18.34	86.49	50.00	4.47	18.03	4.98	24.50	38.80	6.19	1.77
RH 1599-44 × RH 1222-28	42.00	155.67	211.17	8.00	22.17	84.67	61.00	4.20	16.87	4.77	28.55	39.80	9.44	2.83
RH 1599-44 × DRMR 2035	41.00	152.00	199.98	7.50	20.50	76.17	50.67	3.83	17.23	5.16	30.51	40.23	8.89	2.73
RH 1599-44 × EC 597328	43.00	151.33	213.50	6.67	18.33	89.67	60.50	4.27	14.73	4.31	30.10	39.93	9.31	3.07
RH 1599-44 × EC 597317	40.00	145.00	202.00	7.50	18.67	75.84	47.83	4.30	16.17	5.49	30.60	38.73	10.23	3.20
RH 1633 × RH 1222-28	49.00	153.67	211.67	6.50	17.84	76.50	48.33	4.74	15.17	6.92	26.60	38.83	12.21	3.50
RH 1633 × DRMR 2035	45.00	155.33	210.33	7.67	19.50	76.17	54.17	4.57	16.47	6.31	26.40	38.87	9.93	3.10
RH 1633 × EC 597328	45.00	150.33	198.67	6.17	17.34	96.17	60.50	4.90	16.84	5.21	25.05	38.77	13.96	3.57
RH 1633 × EC 597317	46.00	154.00	209.50	7.50	18.17	85.34	59.17	4.83	13.70	6.72	27.20	38.70	12.82	3.47
RH 1657 × RH 1222-28	40.00	149.33	206.71	8.17	22.00	77.83	41.84	4.50	15.57	5.20	21.90	39.87	7.60	2.47
RH 1657 × DRMR 2035	40.00	149.00	194.34	7.67	19.84	85.00	54.00	4.47	15.60	4.98	25.80	38.80	9.24	2.90
RH 1657 × EC 597328	40.00	143.33	209.67	6.84	23.50	90.33	63.00	4.30	14.50	4.47	22.60	38.97	7.68	2.43
RH 1657 × EC 597317	38.00	146.00	211.50	5.83	15.67	72.67	42.17	4.50	16.47	4.66	21.30	38.70	6.37	1.87
RH 1658 × RH 1222-28	36.00	146.00	193.34	7.17	18.50	98.00	55.67	4.83	15.80	6.18	29.80	38.83	9.29	3.03
RH 1658 × DRMR 2035	35.00	147.33	187.17	6.34	17.17	77.00	46.50	4.87	16.50	5.71	20.10	38.87	11.23	3.27
RH 1658 × EC 597328	37.00	150.33	195.83	8.17	23.50	89.84	58.17	4.83	16.27	5.42	31.40	39.90	10.21	2.97
RH 1658 × EC 597317	38.00	146.00	201.50	6.50	20.84	78.67	50.84	4.40	15.57	6.01	23.00	39.33	11.08	3.50
RH 1664 × RH 1222-28	40.00	152.00	204.83	8.34	23.00	87.17	60.67	4.77	16.24	5.78	30.80	38.77	8.41	2.70
RH 1664 × DRMR 2035	39.00	150.33	194.67	7.34	20.67	80.67	51.00	4.73	16.20	5.22	24.70	38.73	8.99	2.57
RH 1664 × EC 597328	38.00	148.33	203.33	8.00	21.00	78.33	63.67	4.27	16.83	4.86	26.30	38.67	11.44	3.57
RH 1664 × EC 597317	42.00	149.00	212.17	6.84	17.34	76.50	53.50	4.20	15.67	4.28	25.70	38.90	13.37	3.83
RH 1899-53 × RH 1222-28	45.00	149.67	215.84	5.84	16.34	72.67	60.84	4.37	15.03	5.22	20.50	38.77	7.85	2.53
RH 1899-53 × DRMR 2035	43.00	151.00	208.34	7.50	22.33	86.33	51.34	4.23	15.80	5.17	30.10	38.83	8.30	2.50
RH 1899-53 × EC 597328	39.00	154.00	217.83	5.83	19.67	89.67	61.84	4.27	17.17	4.89	27.05	38.83	9.74	3.17
RH 1899-53 × EC 597317	47.00	155.00	213.03	6.84	19.16	81.17	56.67	4.40	14.93	5.22	24.90	38.83	10.49	3.23
PM 26 × RH 1222-28	40.00	151.33	194.48	6.67	18.47	84.50	58.17	4.54	17.25	5.18	21.90	38.47	7.94	2.60
PM 26 × DRMR 2035	38.00	149.33	188.17	7.50	25.50	89.17	54.67	4.57	15.57	5.25	29.80	40.07	8.36	2.53
PM 26 × EC 597328	41.00	143.33	197.67	6.67	21.67	88.50	56.83	4.27	15.67	4.87	25.10	38.70	9.98	3.10
PM 26 × EC 597317	36.00	149.00	193.67	5.84	16.34	81.34	50.00	4.34	17.17	4.96	28.35	39.90	11.16	3.40
RH 725 (Check)	46.00	153.67	219.17	7.17	17.00	79.83	58.67	4.75	14.73	6.26	27.85	39.97	9.34	3.43
RH 0749 (Check)	46.00	156.67	215.17	6.00	16.33	79.17	51.17	4.30	15.77	6.79	26.50	38.80	11.39	3.00
S.E. (m)	1.09	3.08	6.23	0.31	0.91	2.41	2.11	0.14	0.50	0.10	2.13	0.07	0.97	0.27
C.D. (*p* ≤ 0.05)	3.06	8.64	17.47	0.88	2.55	6.74	5.92	0.41	1.41	0.27	5.97	0.20	2.72	0.76

DF—Days to 50% flowering; DM—Days to maturity; PH—Plant height (cm); NPB—Number of primary branches/plant; NSB—No. of secondary branches/plant; MSL—Main shoot length (cm); NSMS—No. of siliquae on main shoot; SL—Siliqua length (cm); NSS—No. of seeds/siliqua; TSW—1000-seed weight (g); SYP—Seed yield/plant (g); OC—Oil content (%); MLL—Mean lesion length (cm); DSI—Disease severity index.

**Table 4 plants-11-00671-t004:** General combining ability (GCA) of parents for seed yield and its component traits and Sclerotinia stem rot resistance in Indian mustard.

Parents (Lines + Testers)		DF	DM	PH	NPB	NSB	MSL	NSMS	SL	NSS	TSW	SYP	OC	MLL	DSI
Lines	RH 1566	5.43 **	3.49 *	6.80 *	−0.24	−1.00 *	−4.68 **	0.85	−0.33 **	−0.95 **	−0.09	−0.49	0.51 **	−0.64	−0.06
	RH 1569	4.43 **	0.83	7.63 *	0.96 **	2.54 **	1.39	5.54 **	0.11	−0.09	0.22 **	2.40 *	−0.02	−2.33 **	−0.69 **
	RH 1599-41	−4.58 **	−4.68 **	−1.49	−0.53 **	−1.46 **	1.55	−1.00	−0.29 **	0.82 **	−0.42 **	−1.41	−0.33 **	−2.17 **	−0.71 **
	RH 1599-44	−0.08	1.16	1.21	0.34 *	0.08	−1.23	−0.54	−0.25 **	0.32	−0.33 **	3.58 **	0.56 **	0.36	0.18
	RH 1633	4.68 **	3.49 *	2.09	−0.12	−1.62 **	0.72	0.00	0.36 **	−0.39	1.03 **	−0.05	−0.32 **	3.13 **	0.63 **
	RH 1657	−2.08 **	−2.93 *	0.10	0.05	0.42	−1.36	−5.29 **	0.04	−0.40	−0.43 **	−3.46 **	−0.03	−1.38 **	−0.36 **
	RH 1658	−5.08 **	−2.43	−10.99 **	−0.03	0.17	3.06 *	−2.75 *	0.33 **	0.10	0.57 **	−0.29	0.12 **	1.35 **	0.41 **
	RH 1664	−1.83 **	0.08	−1.70	0.55 **	0.67	−2.15	1.67	0.09	0.30	−0.23 **	0.51	−0.35 **	1.45 **	0.39 **
	RH 1899-53	1.93 **	2.58	8.31 *	−0.58 **	−0.46	−0.36	2.13	−0.08	−0.20	−0.13 **	−0.72	−0.30 **	−0.01	0.08
	PM 26	−2.83 **	−1.59	−11.96 **	−0.41 *	0.66	3.06 *	−0.62	0.03	0.48 *	−0.19 **	−0.07	0.17 **	0.25	0.13
	SE (±) Lines	0.47	1.45	3.22	0.17	0.43	1.20	1.16	0.06	0.20	0.05	1.16	0.04	0.48	0.14
Testers	RH 1222-28	0.33	1.06	−0.10	0.26 **	0.23	0.22	0.39	0.07 *	0.10	0.34 **	0.05	−0.16 **	−0.79 **	−0.17 *
	DRMR 2035	−1.08 **	0.49	−3.84 *	0.19 *	0.50 *	−2.04 **	−2.26 **	−0.03	0.20	0.07 **	0.42	0.22 **	−0.62*	−0.22 **
	EC 597328	−0.08	−1.88 *	2.00	−0.07	0.93 **	4.62 **	5.02 **	−0.10 **	−0.09	−0.51 **	0.13	−0.01	0.72 **	0.21 **
	EC 597317	0.83 **	0.33	1.94	−0.38 **	−1.66 **	2.80 **	−3.16 **	0.06	−0.21	0.10 **	−0.60	−0.05 *	0.69 **	0.18 *
	SE (±) Testers	0.30	0.92	2.04	0.11	0.27	0.76	0.73	0.04	0.13	0.03	0.73	0.02	0.31	0.09

* Significant at *p* ≤ 0.05 and ** Significant at *p* ≤ 0.01. DF-Days to 50% flowering, DM—Days to maturity, PH—Plant height (cm), NPB—Number of primary branches/plant, NSB—Number of secondary branches/plant, MSL—Main shoot length (cm), NSMS—Number of siliquae on main shoot, SL—Siliqua length (cm), NSS—Number of seeds/siliqua, TSW—1000 seed weight (g), SYP—Seed yield/plant (g), OC—Oil content (%), MLL—Mean lesion length (cm), DSI—Disease severity index.

**Table 5 plants-11-00671-t005:** Specific combining ability (SCA) of different crosses for seed yield and its component traits and Sclerotinia stem rot resistance in Indian mustard.

Sr. No.	Crosses	DF	DM	PH	NPB	NSB	MSL	NSMS	SL	NSS	TSW	SYP	OC	MLL	DSI
1.	RH 1566 × RH 1222-28	−1.33	0.61	5.68	0.57	0.43	−1.02	1.21	0.23	0.88 *	0.16	1.88	−0.80 **	0.71	0.19
2.	RH 1566 × DRMR 2035	−0.93	1.51	2.42	0.14	0.66	1.40	1.86	0.10	0.64	0.17	1.01	0.22 **	0.30	0.14
3.	RH 1566 × EC 597328	0.08	−5.46	−2.92	0.59	0.89	−9.03 **	−7.01 **	−0.06	−1.34 **	−0.50 **	−3.31	0.22 **	0.44	0.10
4.	RH 1566 × EC 597317	2.18 *	3.34	−5.19	−1.30 **	−1.98 *	8.66 **	3.93	−0.26 *	−0.18	0.17	0.43	0.36 **	−1.45	−0.43
5.	RH 1569 × RH 1222-28	−1.33	−0.73	−8.15	0.04	2.06 *	4.08	−1.48	−0.15	0.09	−0.09	0.74	−0.20 **	0.51	0.05
6.	RH 1569 × DRMR 2035	−1.93 *	0.51	9.42	−0.40	−1.54	−1.51	−0.49	−0.11	0.32	−0.07	−0.18	0.72 **	−0.49	−0.30
7.	RH 1569 × EC 597328	−0.93	−0.46	−0.08	−0.80 *	−1.81 *	0.67	3.40	−0.50 **	0.29	−0.10	1.25	−0.32 **	−1.08	−0.28
8.	RH 1569 × EC 597317	4.18 **	0.68	−1.19	1.16 **	1.29	−3.24	−1.43	0.76 **	−0.70	0.26 **	−1.81	−0.21 **	1.06	0.53
9.	RH 1599-41 × RH 1222-28	−1.33	−0.89	−11.03	−0.14	−0.44	−1.42	−0.10	−0.19	−0.45	0.17	1.70	0.18 *	−0.61	−0.20
10.	RH 1599-41 × DRMR 2035	0.08	−3.66	3.54	−0.57	−2.38 **	−3.17	3.88	−0.15	−1.18 **	−0.31 **	−1.27	−0.17 *	−0.17	−0.08
11.	RH 1599-41 × 597328	3.08 **	1.04	5.71	0.70 *	1.19	−0.33	−2.39	0.06	0.14	0.09	−0.58	−0.08	2.21 *	0.75 **
12.	RH 1599-41 × EC 597317	−1.83	3.51	1.77	0.01	1.62	4.92 *	−1.39	0.29 *	1.49 **	0.05	0.15	0.07	−1.44	−0.48
13.	RH 1599-44 × RH 1222-28	0.18	3.61	4.60	0.33	2.02 *	2.87	5.61 *	−0.02	0.52	−0.50 **	−1.44	0.29 **	0.76	0.05
14.	RH 1599-44 × DRMR 2035	0.58	0.51	−2.85	−0.11	0.08	−3.38	−2.08	−0.29 *	0.78	0.15	0.15	0.34 **	0.04	−0.00
15.	RH 1599-44 × EC 597328	1.58	2.21	4.84	−0.68 *	−2.52 **	3.46	0.48	0.22	−1.43 **	−0.11	0.03	0.27 **	−0.88	−0.11
16.	RH 1599-44 × EC 597317	−2.33 *	−6.33 *	−6.60	0.46	0.42	−2.95	−4.01	0.09	0.13	0.46 **	1.26	−0.89 **	0.08	0.07
17.	RH 1633 × RH 1222-28	2.43 *	−0.73	4.22	−0.72 *	−0.61	−7.26 **	−7.61 **	−0.10	−0.47	0.29 **	0.24	0.20 **	0.76	0.26
18.	RH 1633 × DRMR 2035	−0.18	1.51	6.63	0.52	0.79	−5.34 *	0.88	−0.16	0.72	−0.05	−0.33	−0.14	−1.69	−0.09
19.	RH 1633 × EC 597328	−1.18	−1.13	−10.87	−0.72 *	−1.81 *	8.00 **	−0.06	0.25 *	1.39 **	−0.56 **	−1.40	−0.02	1.02	−0.06
20.	RH 1633 × EC 597317	−1.08	0.34	0.02	0.92 **	1.62	4.60	6.78 **	0.01	−1.64 **	0.33 **	1.49	−0.04	−0.10	−0.12
21.	RH 1657 × RH 1222-28	0.18	1.36	1.25	0.78 *	1.52	−3.84	−8.81 **	−0.01	−0.07	0.03	−1.05	0.95 **	0.67	0.22
22.	RH 1657 × DRMR 2035	1.58	1.59	−7.38	0.35	−0.91	5.58 *	6.00 *	0.06	−0.134	0.08	2.48	−0.50 **	2.13 *	0.71 *
23.	RH 1657 × EC 597328	0.58	−1.71	2.12	−0.22	2.32 **	4.25	7.73 **	−0.04	−0.94 *	0.16	−0.43	−0.11	−0.76	−0.20
24.	RH 1657 × EC 597317	−2.33 *	−1.24	4.01	−0.92 **	−2.92 **	−5.99 *	−4.93 *	−0.01	1.14 **	−0.27 **	−1.00	−0.34 **	−2.04 *	−0.73 **
25.	RH 1658 × RH 1222-28	−0.83	−2.48	−1.03	−0.13	−1.73 *	11.91 **	2.48	0.03	−0.34	0.01	3.68	−0.24 **	−0.37	0.01
26.	RH 1658 × DRMR 2035	−0.43	−0.58	−3.45	−0.90 **	−3.33 **	−6.84 **	−4.04	0.17	0.27	−0.19	−6.40 **	−0.59 **	1.39	0.30
27.	RH 1658 × EC 597328	0.58	4.79	−0.63	1.20 **	2.57 **	−0.66	0.36	0.20	0.33	0.10	5.19 *	0.68 **	−0.96	−0.44
28.	RH 1658 × EC 597317	0.68	−1.74	5.11	−0.17	2.50 **	−4.41	1.20	−0.39 **	−0.26	0.08	−2.47	0.15 *	−0.06	0.13
29.	RH 1664 × RH 1222-28	−0.08	1.03	1.18	0.45	2.27 **	6.29 *	3.06	0.20	−0.10	0.41 **	3.88	0.16 *	−1.35	−0.30
30.	RH 1664 × DRMR 2035	0.33	−0.08	−5.25	−0.48	−0.33	2.04	−3.95	0.27 *	−0.23	0.11	−2.60	−0.25 **	−0.95	−0.38
31.	RH 1664 × EC 597328	−1.68	0.29	−2.42	0.45	−0.43	−6.96 **	1.44	−0.12	0.69	0.34 **	−0.71	−0.09	0.17	0.18
32.	RH 1664 × EC 597317	1.43	−1.24	6.48	−0.42	−1.50	−1.37	−0.55	−0.35 **	−0.36	−0.85 **	−0.57	0.18 *	2.13 *	0.49
33.	RH 1899-53 × RH 1222-28	1.18	−3.81	2.18	−0.92 **	−3.27 **	−10.00 **	2.77	−0.02	−0.81	−0.25 *	−5.19 *	0.11	−0.46	−0.15
34.	RH 1899-53 × DRMR 2035	0.58	−1.91	−1.59	0.81 *	2.46 **	5.91 *	−4.08	−0.06	−0.14	−0.02	4.04	−0.20 **	−0.18	−0.14
35.	RH 1899-53 × EC 597328	−4.43 **	3.46	2.07	−0.60	−0.64	2.59	−0.85	0.05	1.53 **	0.28 **	1.28	0.03	−0.07	0.09
36.	RH 1899-53 × EC 597317	2.68 **	2.26	−2.67	0.71 *	1.45	1.51	2.16	0.02	−0.59	−0.01	−0.14	0.07	0.71	0.20
37.	PM 26 × RH 1222-28	0.93	2.03	1.08	−0.26	−2.26 **	−1.59	2.86	0.04	0.74	−0.23 *	−4.44	−0.66 **	−0.63	−0.14
38.	PM 26 × DRMR 2035	0.33	0.59	−1.49	0.64	4.51 **	5.33 *	2.01	0.17	−1.05 *	0.12	3.09	0.57 **	−0.39	−0.15
39.	PM 26 × EC 597328	2.33 *	−3.04	2.17	0.08	0.24	−2.00	−3.10	−0.06	−0.65	0.32 **	−1.32	−0.58 **	−0.10	−0.03
40.	PM 26 × EC 597317	−3.58 **	0.43	−1.76	−0.46	−2.49 **	−1.74	−1.76	−0.15	0.96 *	−0.21 *	2.66	0.67 **	1.11	0.32
	SE (±) SCA	0.94	2.89	6.43	0.33	0.85	2.41	2.32	0.12	0.41	0.10	2.31	0.07	0.97	0.27

* Significant at *p* ≤ 0.05 and ** Significant at *p* ≤ 0.01. DF—Days to 50% flowering, DM—Days to maturity, PH—Plant height (cm), NPB—Number of primary branches/plant, NSB—Number of secondary branches/plant, MSL—Main shoot length (cm), NSMS—Number of siliquae on main shoot, SL—Siliqua length (cm), NSS—Number of seeds/siliqua, TSW—1000 seed weight (g), SYP—Seed yield/plant (g), OC—Oil content (%), MLL—Mean lesion length (cm), DSI—Disease severity index.

**Table 6 plants-11-00671-t006:** Contribution of lines, testers, and their interaction to the total variance; GCA, SCA, and GCA/SCA variance, genotypic coefficient of variation (GCV), phenotypic coefficient of variation (PCV), broad sense heritability (h^2^bs), and genetic advance as per cent of mean (GAM)for seed yield and its component traits and Sclerotinia stem rot resistance in Indian mustard.

Characters	Due to Lines (%)	Due to Testers (%)	Due to Lines × Testers (%)	σ^2^ GCA	σ^2^ SCA	σ^2^ GCA/σ^2^ SCA	GCV	PCV	h^2^bs	GAM
DF	79.15	2.82	18.03	4.68 **	3.73 **	1.26	10.84	11.71	85.69	20.66
DM	50.85	8.81	40.34	2.27 **	0.07	32.65	2.07	4.12	25.31	2.15
PH	61.48	7.70	30.82	13.66 **	−8.11	−1.69	5.66	7.72	53.68	8.54
NPB	32.18	9.11	58.71	0.11 *	0.48 **	0.23	11.14	13.61	67.04	18.79
NSB	22.49	16.39	61.12	1.26 *	4.72 **	0.27	11.68	14.18	67.87	19.82
MSL	13.70	21.34	64.97	8.82 *	31.87 **	0.28	7.77	9.31	69.62	13.36
NSMS	23.81	31.64	44.55	11.26 **	15.68 **	0.72	8.66	10.92	62.94	14.15
SL	49.44	4.97	45.59	0.02 **	0.06 **	0.35	8.72	10.5	68.93	14.91
NSS	26.25	2.86	70.89	0.08	0.77 **	0.10	6.76	8.78	59.28	10.72
TSW	53.54	26.10	20.36	0.16 **	0.11 **	1.50	13.3	13.67	94.71	26.67
SYP	35.84	1.49	62.68	0.44	3.37	0.13	11.9	18.87	39.73	15.45
OC	36.09	6.60	57.31	0.05 *	0.24 **	0.21	1.48	1.51	95.55	2.98
MLL	63.18	11.93	24.89	1.18 **	0.60 *	1.97	25.39	31.48	65.05	42.19
DSI	58.45	12.15	29.40	0.09 **	0.07 *	1.29	21.56	25.81	78.07	46.14

* Significant at *p* ≤ 0.05 and ** Significant at *p* ≤ 0.01; DF—Days to 50% flowering, DM—Days to maturity, PH—Plant height (cm), NPB—Number of primary branches/plant, NSB—Number of secondary branches/plant, MSL—Main shoot length (cm), NSMS—Number of siliquae on main shoot, SL—Siliqua length (cm), NSS—Number of seeds/siliqua, TSW—1000 seed weight (g), SYP—Seed yield/plant (g), OC—Oil content (%), MLL—Mean lesion length (cm), DSI—Disease severity index.

**Table 7 plants-11-00671-t007:** Phenotypic (above diagonal) and genotypic (below diagonal) correlation coefficients among Sclerotinia stem rot resistance assessment parameters as well as yield and its component traits in Indian mustard.

Traits	DF	DM	PH	NPB	NSB	MSL	NSMS	SL	NSS	TSW	OC	MLL	DSI	SYP
DF	1.000	0.408 **	0.549 **	0.180 *	−0.027 ^NS^	−0.375 **	0.162 *	−0.144 ^NS^	−0.497 **	0.151 ^NS^	−0.033 ^NS^	−0.007 ^NS^	−0.170 *	−0.113 ^NS^
DM	0.826 **	1.000	0.439 **	0.156 *	0.019 ^NS^	−0.067 ^NS^	0.124 ^NS^	0.123 ^NS^	0.049 ^NS^	0.319 **	0.017 ^NS^	0.099 ^NS^	−0.072 ^NS^	0.175 *
PH	0.754 **	0.804 **	1.000	0.243 **	0.016 ^NS^	−0.196 *	0.228 **	−0.114 ^NS^	−0.201 **	0.100 ^NS^	0.076 ^NS^	−0.114 ^NS^	−0.219 **	0.133 ^NS^
NPB	0.230 **	0.333 **	0.329 **	1.000	0.726 **	0.048 ^NS^	0.272 **	0.231 **	0.009 ^NS^	0.110 ^NS^	−0.076 ^NS^	−0.015 ^NS^	−0.140 ^NS^	0.425 **
NSB	−0.019 ^NS^	0.051 ^NS^	0.037 ^NS^	0.688 **	1.000	0.269 **	0.284 **	−0.019 ^NS^	−0.029 ^NS^	−0.127 ^NS^	−0.130 ^NS^	0.009 ^NS^	−0.189 *	0.343 **
MSL	−0.417 **	−0.244 **	−0.270 **	−0.033 ^NS^	0.270 **	1.000	0.454 **	0.240 **	0.316 **	0.017 ^NS^	−0.209 **	0.005 ^NS^	0.106 ^NS^	0.337 **
NSMS	0.282 **	0.237 **	0.400 **	0.251 **	0.288 **	0.382 **	1.000	−0.041 ^NS^	−0.062 ^NS^	−0.097 ^NS^	−0.027 ^NS^	−0.101 ^NS^	−0.088 ^NS^	0.223 **
SL	−0.109 ^NS^	0.041 ^NS^	0.024 ^NS^	0.263 **	−0.080 ^NS^	0.237 **	−0.108 ^NS^	1.000	0.439 **	0.490 **	−0.061 ^NS^	0.163 *	0.264 **	0.199 **
NSPS	−0.691 **	−0.310 **	−0.258 **	−0.040 ^NS^	−0.068 ^NS^	0.438 **	−0.078 ^NS^	0.376 **	1.000	0.121 ^NS^	0.068 ^NS^	0.004 ^NS^	0.154 *	0.266 **
TSW	0.171 *	0.538 **	0.083 ^NS^	0.130 ^NS^	−0.166 *	0.010 ^NS^	−0.132 ^NS^	0.593 **	0.127 ^NS^	1.000	−0.157 *	0.126 ^NS^	0.148 ^NS^	0.274 **
OC	−0.048 ^NS^	0.045 ^NS^	0.111 ^NS^	−0.087 ^NS^	−0.150 ^NS^	−0.241 **	−0.011 ^NS^	−0.076 ^NS^	0.098 ^NS^	−0.168 *	1.000	−0.055 ^NS^	0.107 ^NS^	−0.061 ^NS^
MLL	−0.001 ^NS^	0.041 ^NS^	−0.083 ^NS^	0.015 ^NS^	−0.025 ^NS^	0.030 ^NS^	−0.095 ^NS^	0.198 *	−0.085 ^NS^	0.139 ^NS^	−0.072 ^NS^	1.000	0.259 **	−0.017 ^NS^
DSI	−0.190 *	−0.128 ^NS^	−0.317 **	−0.167 *	−0.228 **	0.132 ^NS^	−0.094 ^NS^	0.317 **	0.204 **	0.157 *	0.109 ^NS^	0.350 **	1.000	−0.144 ^NS^
SYP	−0.165 *	0.436 **	−0.002 ^NS^	0.458 **	0.278 **	0.551 **	0.335 **	0.425 **	0.597 **	0.311 **	−0.073 ^NS^	0.000 ^NS^	−0.246 **	1.000

* Significant at *p* ≤ 0.05 and ** Significant at *p* ≤ 0.01; ^NS^—Non-significant; DF-Days to 50% flowering; DM—Days to maturity; PH—Plant height (cm); NPB—No. of primary branches/plant; NSB—No. of secondary branches/plant; MSL—Main shoot length (cm), NSMS—No. of siliquae on main shoot; SL—Siliqua length (cm); NSS—No. of seeds/siliqua; TSW—1000-seed weight (g); SYP—Seed yield/plant (g); OC—Oil content (%), MLL—Mean lesion length (cm), DSI—Disease severity index.

**Table 8 plants-11-00671-t008:** Path coefficient based on genotypic correlation analysis for Sclerotinia stem rot resistance assessment parameters and yield component traits indicating direct effect (diagonal and bold) and indirect effect (above and below diagonal) on seed yield in Indian mustard.

Traits	DF	DM	PH	NPB	NSB	MSL	NSMS	SL	NSS	TSW	OC	MLL	DSI
DF	**2.645**	−3.101	1.238	0.275	−0.001	−0.597	−0.210	0.228	−0.931	0.432	−0.027	0.000	−0.116
DM	2.186	**−3.753**	1.319	0.399	0.002	−0.349	−0.176	−0.085	−0.419	1.359	0.025	0.007	−0.078
PH	1.995	−3.016	**1.642**	0.394	0.001	−0.386	−0.298	−0.051	−0.348	0.210	0.063	−0.014	−0.194
NPB	0.609	−1.251	0.540	**1.197**	0.023	−0.048	−0.187	−0.551	−0.053	0.329	−0.049	0.002	−0.102
NSB	−0.050	−0.190	0.061	0.823	**0.034**	0.385	−0.215	0.168	−0.092	−0.419	−0.084	−0.004	−0.139
MSL	−1.104	0.916	−0.443	−0.040	0.009	**1.430**	−0.285	−0.496	0.590	0.024	−0.136	0.005	0.081
NSMS	0.747	−0.888	0.656	0.300	0.010	0.547	**−0.745**	0.226	−0.105	−0.332	−0.006	−0.016	−0.058
SL	−0.289	−0.153	0.040	0.315	−0.003	0.339	0.080	**−2.092**	0.506	1.497	−0.043	0.033	0.194
NSS	−1.827	1.165	−0.423	−0.047	−0.002	0.626	0.058	−0.786	**1.349**	0.320	0.055	−0.014	0.124
TSW	0.453	−2.020	0.136	0.156	−0.006	0.014	0.098	−1.240	0.171	**2.525**	−0.094	0.023	0.096
OC	−0.127	−0.169	0.183	−0.105	−0.005	−0.345	0.008	0.159	0.132	−0.423	**0.564**	−0.012	0.067
MLL	−0.003	−0.153	−0.136	0.018	−0.001	0.043	0.071	−0.414	−0.115	0.351	−0.041	**0.167**	0.214
DSI	−0.503	0.479	−0.520	−0.200	−0.008	0.189	0.070	−0.663	0.275	0.396	0.062	0.058	**0.611**

Residual are 0.1172; DF—Days to 50% flowering; DM—Days to maturity; PH—Plant height (cm); NPB—No. of primary branches/plant; NSB—No. of secondary branches/plant; MSL—Main shoot length (cm), NSMS—No. of siliquae on main shoot; SL—Siliqua length (cm); NSS—No. of seeds/siliqua; TSW—1000-Seed weight (g); SYP—Seed yield/plant (g); OC—Oil content (%), MLL—Mean lesion length (cm), DSI—Disease severity index.

## Data Availability

Data will be available on the basis of reasonable request to corresponding authors.

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
