# Peer review of "Genetic Analysis for Resistance to Sclerotinia Stem Rot, Yield and Its Component Traits in Indian Mustard [Brassica juncea (L.) Czern & Coss.]"

_plants, 2022, doi:10.3390/plants11050671_

Round 1
Reviewer 1 Report
This submission from Singh and colleagues is a revised version of a prior submission. The manuscript describes genetic analysis of Sclerotinia stem rot resistance as well as yield and related traits in the oilseed crop Brassica juncea (Indian mustard). The authors carry out a line x tester crossing scheme to evaluate general combining ability and specific combining ability among 10 lines and 4 testers for Sclerotinia resistance, yield, and a number of specific traits contributing to yield. These results should be useful in B. juncea breeding efforts. The analyses are suitable, and the presentation of the results is appropriate. The English language writing is substantially improved compared to the prior submission. I have a few comments, detailed below:
- I raised this point in the previous review, but the authors do not seem to have addressed it. If I understand correctly from the methods section, the yield and other associated characteristics were measured from 10 plants per genotype/replication that had been inoculated with S. sclerotiorum and apparently were not also measured from uninoculated plants. Is this accurate? If this is the case, the authors should make this clear in the discussion and point out that the measured yields and characteristics reflect performance after Sclerotinia inoculation and do not necessarily reflect performance of these lines or combinations in the absence of disease pressure. If my understanding of the methods is not correct, the authors should clarify this in the description of the study design.
- The discussion has been greatly expanded from the prior submission and is now excessively long with unnecessary repetition of the results. The authors should reduce the length of the discussion section, avoid lengthy repetition of results, and make the discussion more focused on key findings.
Author Response
Minor Revisions: Response to Reviewer reports
Manuscript ID: plants-1564758: “Genetic analysis for resistance to Sclerotinia stems rot, yield and its component traits in Indian mustard [Brassica juncea (L.) Czern &
Coss.]”
We are grateful to the Editor and reviewers for their insightful comments and helpful suggestions on the paper. We amended the paper in response to the comments, and the following is a point-by-point response to the questions:
Reviewer #1
Comment 1: I raised this point in the previous review, but the authors do not seem to have addressed it. If I understand correctly from the methods section, the yield and other associated characteristics were measured from 10 plants per genotype/replication that had been inoculated with S. sclerotiorum and apparently were not also measured from uninoculated plants. Is this accurate? If this is the case, the authors should make this clear in the discussion and point out that the measured yields and characteristics reflect performance after Sclerotinia inoculation and do not necessarily reflect performance of these lines or combinations in the absence of disease pressure. If my understanding of the methods is not correct, the authors should clarify this in the description of the study design.
Response:
We appreciate the first-round comments from the reviewer, which helped us improve the manuscript. The yield and other associated characteristics were measured in 10 un-inoculated and healthy plants per genotype/replication. In contrast, disease data were measured in 10 other plants (other than 10 representative plants selected for the assessment of yield and its component traits) inoculated with S. sclerotiorum per genotype/replication. The reason for using uninoculated plants to measure yield and its component traits is because S. sclerotiorum infection on sensitive hosts/genotypes causes stem collapse and plant lodging, both of which result in significant reductions in seed yield (up to 90%) and quality. On the other hand, resistant hosts/genotypes have a minor impact on the yield. Genetic resistance has been the foundation of disease control for many crop species to reduce yield losses. As a result, the primary goal of this study was to conduct a genetic analysis of Sclerotinia stem rot resistance and yield and yield component traits to combine resistance with high yield potential. However, your idea to evaluate the influence of Sclerotinia stem rot on yield and its component qualities by assessing yield and its component traits on both infected and non-inoculated plants is commendable and will be incorporated in future trials. As per your kind suggestion, we clarified this in the description of the study design as:
2.2. Field evaluation and data collection for seed yield and its component traits
The seed yield and its component traits were recorded as follow: days to 50% flowering (number of days from sowing to when 50% of the plants had flowered) and days to maturity (number of days from sowing to when 80% of the plants had reached physiological maturity) both recorded on plot basis, while other phenotypic traits were recorded from ten randomly selected un-inoculated/healthy plants from the centre of each row as follows: plant height (measured in cm from the base of the plant to tip of the main raceme), number of primary branches/plant (assessed branches derived from the base of the main stem); number of secondary branches/plant (assessed branches emerged from the primary branches), main shoot length (measure in cm from base of most top primary branch to the tip of the plant), number of siliquae on main shoot (siliquae borne on main raceme), siliqua length (measured in cm from the 9 individual siliqua/plant, three each from bottom-, middle-, and top-borne branches), number of seeds/siliqua (counted from the same siliquae used to estimate siliqua length), 1000-seed weight (grams), seed yield/plant (grams) and oil content (determined in % using Soxhlet apparatus method).
2.3. Sclerotinia sclerotiorum inoculum preparation, artificial disease inoculation and resistance assessment
For the pure culture preparation, S. sclerotiorum sclerotia (Hisar isolate) were employed. These sclerotia were first precisely washed in double distilled water, then surface sterilized by soaking for 10-12 sec in 0.1% sodium hypochlorite solution. Sclerotia were then gently rinsed 3-4 times with distilled water to remove any remaining germicide before being dehydrated on aseptic blotter paper. Finally, these sclerotia were cut in half using surgical blades and aseptically transferred to Petri-plates containing Potato Dextrose Agar (PDA). These plates were kept in a BOD incubator at 22±1°C for five days. Using the techniques provided by Li et al. [26] and Singh et al. [1] a five-day-old pure culture of S. sclerotiorum was used to inoculate the main stems of 10 randomly chosen and labeled plants (other than the 10 representative plants selected for the assessment of yield and its component traits) from each genotype/replication. Mycelial discs (5 mm2) were cut from the borders of pure culture plates, placed on a paraffin wax strip (together with a moist cotton swab) and wrapped tightly around the main stem, as in our prior investigations [1,6]. Lesion length (cm) was measured on each infected plant using a linear ruler at 20 days after inoculation for each genotype/replicate, and the average was taken. According to the scale proposed by Singh et al. [1], genotypes were classified as highly resistant [mean lesion length (MLL) ≤ 2.5 cm], resistant (2.6-5.0 cm), moderately resistant (5.1-7.5 cm), susceptible (7.6-10.0 cm), and highly susceptible (>10.0 cm) based on MLL.
Comment 2: The discussion has been greatly expanded from the prior submission and is now excessively long with unnecessary repetition of the results. The authors should reduce the length of the discussion section, avoid lengthy repetition of results, and make the discussion more focused on key findings.
Response:
We again thank the learned reviewer for critically evaluating our work. We have significantly shortened the discussion section in response to your helpful recommendation to improve the manuscript.

Reviewer 2 Report
The manuscript has improved a lot from its previous version and has less similarity index with other articles. The authors have taken care of my concern and hence this manuscript can be accepted in its current format.
Author Response
Minor Revisions: Response to Reviewer reports
Manuscript ID: plants-1564758: “Genetic analyses for resistance to Sclerotinia stems rot, yield and its component traits in Indian mustard [Brassica juncea (L.) Czern &
Coss.]”
We are grateful to the Editor and reviewers for their insightful comments and helpful suggestions on the paper. We amended the paper in response to the comments, and the following is a point-by-point response to the questions:
Reviewer #2
Comment 1: The manuscript has improved a lot from its previous version and has less similarity index with other articles. The authors have taken care of my concern and hence this manuscript can be accepted in its current format.
Response:
We thank the reviewer for his/her first-round remarks, which helped us to improve the manuscript so that it could be considered for publication in Plants.

Reviewer 3 Report
In the manuscript "Genetic analyses for resistance to Sclerotinia stem rot, yield and its component traits in Indian mustard [Brassica juncea (L.) Czern & Coss.]", the authors presented a detailed analysis of 14 Indian mustard lines with a focus on providing guidelines for future breeding experiments. For each line, the authors evaluated resistance to sclerotinia stem rot and favourable agroeconomic traits. The authors also produced crosses between these strains and evaluated disease resistance and economic value of the offspring. Using a detailed statistical analysis, the study provides a robust framework for use in future breeding endeavours, while also highlighting the genetic complexities underlying resistance and economic traits, and the difficulty in combining these through breeding. I enjoyed reading the paper, and although the study was large and very technical, yielded a manuscript is presented in an organised and is technically and scientifically sound. The authors should be congratulated for their contribution.
Although plant breeding is not my area of expertise, I can offer some suggestions that would improve the quality of this manuscript.
- The opening paragraph of the introduction is quite long and becomes difficult to follow. I would suggest splitting the paragraph at line 58 (Indian mustard…) to create a new paragraph that deals with Indian mustard cultivation as opposed to the discussion on cultivation of food crops as seen in the first part of the paragraph.
- Section 2.2 on “Sclerotinia sclerotiorum inoculum preparation and stem inoculation” should be moved and incorporated into section 2.3.2. This would improve clarity as that section already deals with Sclerotinia infection studies.
- No information is given in section 2.2 or 2.3.2 on the time that lesion length was measured for the inoculations conducted. This should be provided.
- I would strongly recommend that the S. sclerotiorum strain used for the inoculation study should be deposited in a publicly accessible culture collection. The heart of this study is related to resistance to sclerotinia, and this would be influenced by the choice of pathogen strain. I do note that the authors state that data for this study would be made available upon request, but I still would argue these strains should be easily accessible, and should have a form of culture identification.
- In lines 244-247, I am not sure what those are, or why they are there? This should be corrected.
- In the first paragraph of the discussion (lines 463 – 467), the section deals with the conditions deemed optimal for Sclerotinia infection development. It is also then mentioned that “the crop’s field conditions were favorable for disease growth”. I have two concerns here.
- Firstly, it completely disrupts the information in the paragraph. The part before (lines 446-463) deals with the aims of the study and the major finding, while the second part (lines 467-482) deals with the need to have genetic variation and understand how it is passed on and compounded in the offspring. I would recommend that the authors consider making the last section a new paragraph.
- If the presented information on climatic conditions in the field was pertinent to the study, this should explicitly be stated in the materials and methods or results, and not in the discussion only. However, I do believe the climatic conditions are not valuable to the outcomes of the study, as artificial inoculations were made, and data gathered before full-scale onset of Sclerotinia disease symptoms. Either way, this section should be removed from its current location.
- Several of the earlier paragraphs in the discussion lack proper references despite making sweeping statements. For example, in lines 483-92, statements are made about combining ability and GCA, how it is utilized in plant breeding. However, no references are provided that would allow a reader to qualify these statements. A similar issues arises in lines 505-516.
- In line 584, a reference is made to “prior results”. Was this from this study (then the relevant finding should be mentioned), or from previous work (a reference should be added)?
- The two paragraphs from line 649 to line 666 can be combined into one. They deal with similar subject matter and would be stronger as a single paragraph.
- The conclusion presented in lines 708-722 contains many statements and much information already presented elsewhere. I do not think it adds to the value of the paper and urge the authors to consider its usefulness.
Some smaller editorial comments:
- Sclerotinia stem rot is referred to a Sclerotinia stem rot and sclerotinia stem rot throughout. Although both forms are used in literature, the authors should choose one version (capitalize or non-capitalized) and be consistent throughout.
- The use of analysis and analyses should be checked and corrected (where needed) throughout. One example: In the title Analyses are used, while in the abstract (line 24) analysis is used.
- Line 28: /per. The / and per means the same thing, and one should be removed.
- Line 36: per se should be italics.
- Line 41: two full-stops present.
- Line 48: “De-spite, India is on track”. Despite should be corrected, but this is also likely not the right word choice as the meaning is unclear.
- Line 85: Species name (sclerotiorum) should be lowercase.
- Line 89: survival structure of (the/this) pathogen.
- Line 115: Characteristics should be characteristic.
- Line 118: Apothecia release ascospores, not apothecial.
- Lines 118-119: and get stuck on healthy siliqua.
- Line 195: Stripe should be strip.
- Line 203: I am not sure if a “12 seed yield and its components” refer to a common technique, but this was very unclear. Please elaborate on this.
- Lines 224 and 228, please release “It” at the start of the sentence with the name of the assay.
- The wording in lines 233-235 is confusing. It sounds as if the work was done in a different study (Kempthorne) to this one, but based on another study (Sing).
- Line 258: Add an “and” between MLL and DSI.
- Line 318: parents’ should be parental.
- Line 391: Space needed between “(GAM)for”.
- Line 550: “in the manners of” might be better worded as “with regards to”.
- Line 583: Consider alternative wording, for example: The contribution to the total variation of the lines, testers, and their interactions support…
- Lines 657: Remove “numerous prior research”.
Author Response
Minor Revisions: Response to Reviewer reports
Manuscript ID: plants-1564758: “Genetic analyses for resistance to Sclerotinia stems rot, yield and its component traits in Indian mustard [Brassica juncea (L.) Czern &
Coss.]”
We are grateful to the Editor and reviewers for their insightful comments and helpful suggestions on the paper. We have updated the manuscript to reflect the feedback, and a point-by-point response to the questions is included below:
Reviewer #3
Comment 1: The opening paragraph of the introduction is quite long and becomes difficult to follow. I would suggest splitting the paragraph at line 58 (Indian mustard…) to create a new paragraph that deals with Indian mustard cultivation as opposed to the discussion on cultivation of food crops as seen in the first part of the paragraph.
Response: We thank the learned reviewer for critical comment. As per the suggestion, we have separated the paragraph at line 58.
Comment 2: Section 2.2 on “Sclerotinia sclerotiorum inoculum preparation and stem inoculation” should be moved and incorporated into section 2.3.2. This would improve clarity as that section already deals with Sclerotinia infection studies.
Response: We thank the learned reviewer for critical comment. We have moved the section to the right location in the amended document, as per the suggestion. It now reads as follows:
2.2. Field evaluation and data collection for seed yield and its component traits
2.3. Sclerotinia sclerotiorum inoculum preparation, artificial disease inoculation and resistance assessment
Comment 3: No information is given in section 2.2 or 2.3.2 on the time that lesion length was measured for the inoculations conducted. This should be provided.
Response: We thank the learned reviewer for critically observation. We have measured the lesion length on each inoculated plant at 20 days after inoculation as per the standard protocol adopted in our previous work (Singh et al. 2021; Singh et al. 2020). We have incorporated this information in section 2.3 that reads like:
Lesion length (cm) was measured on each infected plant using a linear ruler at 20 days after inoculation for each genotype/replicate, and the average was taken.
Comment 4: I would strongly recommend that the S. sclerotiorum strain used for the inoculation study should be deposited in a publicly accessible culture collection. The heart of this study is related to resistance to sclerotinia, and this would be influenced by the choice of pathogen strain. I do note that the authors state that data for this study would be made available upon request, but I still would argue these strains should be easily accessible, and should have a form of culture identification.
Response: It has been stated in the manuscript that the data generated during the present study will be made available upon formal request. Your suggestion that the S. sclerotiorum strain utilized in this study be deposited in a publicly accessible culture collection is commendable, and it will be done soon.
Comment 5: In lines 244-247, I am not sure what those are, or why they are there? This should be corrected.
Response: We thank the reviewer for keen observation. These are (lines 244-247) typographical errors and have been removed from the manuscript.
Comment 6: In the first paragraph of the discussion (lines 463 – 467), the section deals with the conditions deemed optimal for Sclerotinia infection development. It is also then mentioned that “the crop’s field conditions were favorable for disease growth”. I have two concerns here.
Comment 6.1: Firstly, it completely disrupts the information in the paragraph. The part before (lines 446-463) deals with the aims of the study and the major finding, while the second part (lines 467-482) deals with the need to have genetic variation and understand how it is passed on and compounded in the offspring. I would recommend that the authors consider making the last section a new paragraph.
Response: We thank the reviewer for suggestions. We have made the last part a new paragraph and moved the portion dealing with the conditions deemed optimal for Sclerotinia infection development to the Materials and Methods Section, as per your suggestions.
Comment 6.2: If the presented information on climatic conditions in the field was pertinent to the study, this should explicitly be stated in the materials and methods or results, and not in the discussion only. However, I do believe the climatic conditions are not valuable to the outcomes of the study, as artificial inoculations were made, and data gathered before full-scale onset of Sclerotinia disease symptoms. Either way, this section should be removed from its current location.
Response: We thank the reviewer for valuable input. The part dealing with the conditions deemed optimal for Sclerotinia infection development has been relocated to the Materials and Methods section of the manuscript, as a result of your excellent comments.
Comment 7: Several of the earlier paragraphs in the discussion lack proper references despite making sweeping statements. For example, in lines 483-92, statements are made about combining ability and GCA, how it is utilized in plant breeding. However, no references are provided that would allow a reader to qualify these statements. A similar issue arises in lines 505-516.
Response: We thank the reviewer for valuable comments. We have included the references (Stuber, 1994 [37]; Fasahat et al. 2016 [38]; Ferreira et al. 2018 [63]; Labroo et al. 2021 [64]) so that the reader can qualify the claims in lines 483-492 and 505-516.
Comment 8: In line 584, a reference is made to “prior results”. Was this from this study (then the relevant finding should be mentioned), or from previous work (a reference should be added)?
Response: We thank the reviewer for keen observation. Previous studies about the importance of general and specific effects for numerous characters, such as Sclerotinia stem rot resistance, yield, and its component qualities, backed up our findings. As a result, the references to these earlier researches have been added to the revised manuscript, as per your recommendation.
Comment 9: The two paragraphs from line 649 to line 666 can be combined into one. They deal with similar subject matter and would be stronger as a single paragraph.
Response: We thank the reviewer for suggestion. Accordingly, we have consolidated these two paragraphs (lines 649 to 666) into one
.
Comment 10: The conclusion presented in lines 708-722 contains many statements and much information already presented elsewhere. I do not think it adds to the value of the paper and urge the authors to consider its usefulness.
Response: We thank the reviewer for valuable suggestion. We have revised the conclusion section as below:
Conclusion:
This study highlighted the rewarding parents and crosses of Indian mustard that can be exploited to launch effective breeding strategies for developing high-yielding and Sclerotinia stem rot-resistant cultivars. Based on results, it is concluded that significant and desirable GCA effects were shown by DRMR 2035 among the testers and RH 1569 among the lines that were considered to be good general combiners for Sclerotinia stem rot resistance, seed yield, and oil content, as well as for most of the yield attributing traits. The cross combination of RH 1657 × EC 597317 and RH 1658 × EC 597328 showed excellent SCA performance for the resistance and yield contributing traits. These parents and hybrids could be utilized as good combiners and isolated to obtain desirable segregate for combining high resistance levels to Sclerotinia stem rot with superior seed yield and oil content in the future Indian mustard breeding program. This study was suggested for a breeding program for which the advantages of both types of variances, namely additive and non-additive, are chosen. Days to 50% flowering, number of primary branches/plant, main shoot length, and 1000-seed weight and disease severity index are the best selection criterion in the field for use in breeding programs for simultaneous improvement in yield and resistance because of its high heritability and desirable association with seed yield.
Some smaller editorial comments:
Comment 1: Sclerotinia stem rot is referred to a Sclerotinia stem rot and sclerotinia stem rot throughout. Although both forms are used in literature, the authors should choose one version (capitalize or non-capitalized) and be consistent throughout.
Response: We thank the reviewer for suggestion. As per your kind suggestion, we have consistently used capitalized version (Sclerotinia stem rot) throughout the manuscript.
Comment 2: The use of analysis and analyses should be checked and corrected (where needed) throughout. One example: In the title Analyses are used, while in the abstract (line 24) analysis is used.
Response: We thank the reviewer for suggestion. The use of analysis and analyses has been checked very carefully and corrected in the revised manuscript.
Comment 3: Line 28: /per. The / and per means the same thing, and one should be removed.
Response: We thank the reviewer for suggestion. We have removed the word “per”.
Comment 4: Line 36: per se should be italics.
Response: Done.
Comment 5: Line 41: two full-stops present.
Response: We have removed the extra dot.
Comment 6: Line 48: “De-spite, India is on track”. Despite should be corrected, but this is also likely not the right word choice as the meaning is unclear.
Response: We thank the reviewer for observation. We have removed this word.
Comment 7: Line 85: Species name (sclerotiorum) should be lowercase.
Response: Done.
Comment 8: Line 89: survival structure of (the/this) pathogen.
Response: Done.
Comment 9: Line 115: Characteristics should be characteristic.
Response: Done.
Comment 10: Line 118: Apothecia release ascospores, not apothecial.
Response: Done.
Comment 11: Lines 118-119: and get stuck on healthy siliqua.
Response: Done
Comment 12: Line 195: Stripe should be strip.
Response: Done.
Comment 13: Line 203: I am not sure if a “12 seed yield and its components” refer to a common technique, but this was very unclear. Please elaborate on this.
Response: Here, the 12 seed yield and its components refer to yield attributing traits viz., seed yield/plant and 11 of its attributing traits including oil content. Changes have been made in the final MS file as per your worthy suggestion.
Comment 14: Lines 224 and 228, please release “It” at the start of the sentence with the name of the assay.
Response: We thank the reviewer for suggestion. As per your suggestions, the word “It” has been replaced with concerned assay name’ in the final manuscript.
.
Comment 15: The wording in lines 233-235 is confusing. It sounds as if the work was done in a different study (Kempthorne) to this one, but based on another study (Singh).
Response: Our investigation was based on Kempthorne's (1957) Line x Tester analysis process, which was further elaborated by Singh and Chaudhary (1985). However, we examined our findings by taking into account both research.
Comment 16: Line 258: Add an “and” between MLL and DSI.
Response: Done.
Comment 17: Line 318: parents’ should be parental.
Response: Done.
Comment 18: Line 391: Space needed between “(GAM)for”.
Response: Done.
Comment 19: Line 550: “in the manners of” might be better worded as “with regards to”.
Response: Done.
Comment 20: Line 583: Consider alternative wording, for example: The contribution to the total variation of the lines, testers, and their interactions support…
Response: Done.
Comment 21: Lines 657: Remove “numerous prior research”.
Response: Done.

Round 2
Reviewer 1 Report
I thank the authors for their attention to revisions. The clarification that yield traits were measured from uninoculated plants is very helpful. At this point, the authors have satisfactorily addressed my comments.
This manuscript is a resubmission of an earlier submission. The following is a list of the peer review reports and author responses from that submission.
Round 1
Reviewer 1 Report
This submission from Singh and colleagues describes genetic analysis of Sclerotinia stem rot resistance as well as yield and related traits in the oilseed crop Brassica juncea (Indian mustard). The authors carry out a line x tester crossing scheme to evaluate general combining ability and specific combining ability among 10 lines and 4 testers for Sclerotinia resistance, yield, and a number of specific traits contributing to yield. These results should be useful in B. juncea breeding efforts. The analyses are suitable, and the presentation of the results is appropriate. However, I think the manuscript can be improved for English language presentation and other specific points detailed below:
1. The manuscript would benefit from English language editing. There are many spelling and grammatical errors, too numerous to list here.
2. The authors should check the formatting of Tables 2 and 3. There are issues with text wrapping and cell size that need to be corrected.
3. If I understand correctly from section 2.3.1, the yield and other associated characteristics were measured from 10 plants per genotype/replication that had been inoculated with S. sclerotiorum and apparently were not also measured from uninoculated plants. Is this accurate? If this is the case, the authors should make this clear in the discussion and point out that the measured yields and characteristics reflect performance after Sclerotinia inoculation and do not necessarily reflect performance of these lines or combinations in the absence of disease pressure.
4. Line 288: “…showed significantly positive and undesirable SCA effect for this trait.” – Specify which trait. I assume DSI/MLL, but it should be clearly indicated.
Reviewer 2 Report
In this manuscript, the author did the genetic analyses for resistance to sclerotinia stem rot, yield, and its component traits in Indian mustard [Brassica juncea (L.) Czern & Coss.]. In this study, the author did genetic analyses using a line × tester matting design by taking 10 susceptible lines and four resistant testers. The significance of GCA and SCA variances indicated the involvement of both additive and non-additive gene action in the inheritance of sclerotinia stem rot resistance as well as for yield attributing traits. The genotype RH 1569 (line) and DRMR 2035 (tester) appeared to be the best general combiners for sclerotinia stem rot resistance in addition to 1000-seed weight, several primary and secondary branches per plant. Among different cross combinations, RH 1657 × EC 597317 was the only cross that showed a significant desirable SCA value for sclerotinia rot resistance. The cross RH 1658 × EC 597328 was best in terms of SCA effects for yield and its component traits and also exhibited, although insignificant, but desirable negative SCA effect for resistance. Based on GCA effects of parents, per se performance, and SCA effects of hybrids, DRMR 2035, RH 1222-28, RH 1569, RH 1599-41, RH 1657, RH 1658, and EC 597328 are promising genotypes to be used as parents in future heterosis breeding and/or for obtaining populations with high yield potential and greater resistance to sclerotinia stem rot in Indian mustard.
The manuscript is written very poorly and needs serious English Editing, and there is no genetic about this study needs morphological analysis and some genetic analysis to be able to at least consider for publication in this journal. Also, the manuscript is heavily plagiarized, as I have mentioned below specifically.
This manuscript is heavily plagiarized at L25-26, L38-39, L40-45, L56-57, L62-64, L75-77, L81-83, L98-99, L100-102, L138-140, L145-146, L151-154, L158-159, L164-165, L220-221, L235-236, L307-310, L316-317, L346-348, L375-376, L384-386, L390-391, L393-394, L395-400, L400-402, L404-405, L436-437, L440-443, L493-496, L497-506, L507-509, L519-522, L523-525, L529-531.